# No Shifted Augmentations (NSA): compact distributions for robust self-supervised Anomaly Detection

## Abstract

Unsupervised Anomaly detection (AD) requires building a notion of normalcy, distinguishing in-distribution (ID) and out-of-distribution (OOD) data, using only available ID samples. Recently, large gains were made on this task for the domain of natural images using self-supervised contrastive feature learning as a first step followed by kNN or traditional one-class classifiers for feature scoring. Learned representations that are non-uniformly distributed on the unit hypersphere have been shown to be beneficial for this task. We go a step further and investigate how the *geometrical compactness* of the ID feature distribution makes isolating and detecting outliers easier, especially in the realistic situation when ID training data is polluted (i.e. ID data contains some OOD data that is used for learning the feature extractor parameters). We propose novel architectural modifications to the self-supervised feature learning step, that enable such compact distributions for ID data to be learned. We show that the proposed modifications can be effectively applied to most existing self-supervised objectives, with large gains in performance. Furthermore, this improved OOD performance is obtained without resorting to tricks such as using strongly augmented ID images (e.g. by 90 degree rotations) as proxies for the unseen OOD data, as these impose overly prescriptive assumptions about ID data and its invariances. We perform extensive studies on benchmark datasets for one-class OOD detection and show state-of-the-art performance in the presence of pollution in the ID data, and comparable performance otherwise. We also propose and extensively evaluate a novel feature scoring technique based on the angular Mahalanobis distance, and propose a simple and novel technique for feature ensembling during evaluation that enables a big boost in performance at nearly zero run-time cost compared to the standard use of model ensembling or test time augmentations. Code for all models and experiments will made open-source.

## 1 Introduction

Anomaly detection (AD) or out-of-distribution (OOD) detection requires using only available in-distribution (ID) samples for training a classifier to decide upon the relative normalcy of samples at test time, without knowledge of the nature of the OOD data. OOD detection is an important problem with practical applications in, for example, industrial defect detection, fraud detection, autonomous driving, biometrics, spoofing detection and many other domains (Tack et al., 2020).

### 1.1 Background

For natural images (which according to the manifold hypothesis lie in a compact set in a suitable space), OOD detection translates to finding as tight a decision boundary as possible around the normal set, while excluding unseen samples from other classes or distributions. This detection was traditionally done with either generative (Zhai et al., 2016) or discriminative models (Schölkopf et al., 2000) on top of shallow features. Deep representations subsequently provided a large boost in performance. However, the density of deep generative image models have often proved to be ineffective (Kingma & Dhariwal, 2018), with poorly calibrated likelihoods away from observed data. There have instead been two recent directions to learn suitable deep features used for OOD evaluation:

a) supervised pre-training on an external dataset. b) Self-supervised learning (SSL) pre-training on either the normal set only or also on an external dataset. A variety of learned metrics, scoring functions, or one-class classifiers have then been employed on top of these learned features and this general paradigm has shown to be highly effective in many cases (Sohn et al., 2020).

The best recent results with SSL-based training for OOD has been with contrastive learning (Tack et al., 2020; Sohn et al., 2020; Sehwag et al., 2021; Winkens et al., 2020). Contrastive learning has been shown to distribute ID data uniformly on the hypersphere (Wang & Isola, 2020). While this helps general multi-class SSL training, it hurts OOD detection, as it makes isolating outliers from the single class harder (Sohn et al., 2020). This uniformity also makes OOD detection much more sensitive to pollution in the inlier training data (Han et al., 2021; Sohn et al., 2020). See Appendix H for more background on related methods.

If some labeled OOD data are available as negatives, semi-supervised learning maybe be used (Ruff et al., 2018; Han et al., 2021). If no such labeled negatives are available, one way to soften the effect of a contrastively learned uniform representation is to introduce artificial negative samples as proxies for these outliers. Using hard augmentations (e.g. 90 degree rotations) has been termed *Distributional Shifting* (Tack et al., 2020; Sohn et al., 2020; Li et al., 2021; Tian et al., 2021). Such augmentations are intended to make the in-distribution data less uniform, and thus easier to isolate from OOD data. However, they also make the significant assumption that the data is not (fully or partially) invariant to those augmentation(s), and that the augmentations are a good proxy for the true negative distribution.

The other direction is using features from a model pre-trained on a large external dataset, with the hope of producing universal features that can work in any OOD detection scenario. This can be done in either supervised (Reiss et al., 2020) or self-supervised manner (Xiao et al., 2021). However, the assumption that such representative labeled samples will be available in the latter case, or that learned image features from general datasets such as ImageNet will transfer well may be restrictive at best.

## 1.2 CONTRIBUTION

In this work, we first investigate the training dynamics of contrastive SSL methods, and show that their performance decays significantly over long term training. We find that uniformity or non-compactness of the learned ID representation is the main reason for this decay. We study this effect on positive-pairs-only SSL using SimSiam (Chen & He, 2020), and show that in such cases, the decay does not happen.

We propose an architectural modification that can be applied generally across such networks, and show extensive analysis that this modification improves performance and always encourages learning a more compact ID representations. In doing so we are able to learn high quality One-class classifiers without resorting to distributionally shifted augmented samples as negatives, hence we term the resulting methodology *No Shifted Augmentations* (NSA). We summarize our contributions as follows:

- We investigate and empirically verify and quantify that the the non-uniformity and compactness of learned ID is a main factor of the final OOD detection performance, independent from the quality of the learned features.

- We propose an assumption-free, simple and novel architectural modification for inducing a non-uniform learned ID representation, and show that this works very well with both SimSiam and SimCLR and produces solid performance improvements.

- We identify, investigate, and solve a gradient problem in SimSiam (and also BYOL) that greatly affects the proper propagation of the norms inside the network; we solve it and notice much higher stability, especially in the low batch size training regime.

- We consider improved feature scoring methods for OOD detection, including in our proposed solution a Mahalanobis Cosine score on nearest neighbors, related to methods in open-set metric learning. We then present a computationally efficient method of feature ensembling that also boosts performance.

- We show unexpected case(s) (e.g. SVHN) where the usual ImageNet-Pretrained ResNet methods fail catastrophically on One-Class Classification Anomaly Detection tasks, even with feature adaptation. We show that training from scratch without using shifted augmentations avoids this.

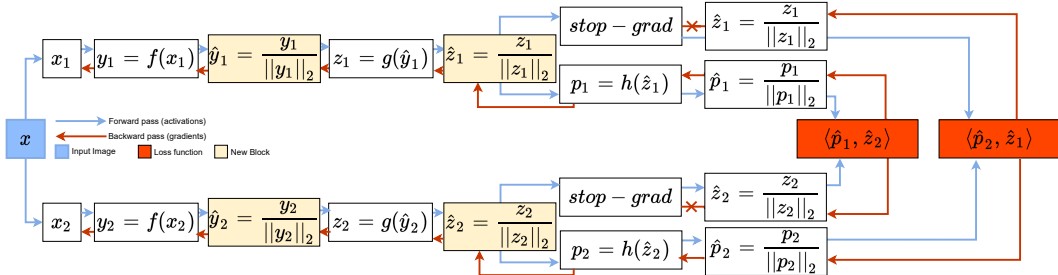

Figure 1: An illustration of modified SimSiam with the yellow boxes showing our proposed changes described in Sec. 2.3, 2.4. In the original version the gradients of $\hat{z}_i = z_i/||z_i||_2$ are prevented from flowing backwards in *all possible paths* by the stop-grad operation. In our modified version, thanks to the added operations, we can mimic the flow of the gradients of $\hat{z}_i = z_i/||z_i||_2$ that were blocked.

- We extensively evaluate and ablate the proposed models with a wide variety of different datasets and scenarios, separating the contributions of representation learning, scoring, data augmentation, and additional variations like ensembling. We show our solutions have comparable performance against more complex methods. More importantly they show state-of-the-art performance, by a wide margin, achieving robustness for small batch sizes and in the presence of polluted data.

## 2 METHODOLOGY

### 2.1 SELF-SUPERVISED LEARNING (SSL) SETUP

A general SSL setup resembling SimSiam (Chen & He, 2020) is depicted in Figure 1.2. SimCLR is a very similar architecture that omits a prediction network, so we use SimSiam for this illustration. The figure shows both the forward pass and backward pass of two random augmentations $x_1$ and $x_2$ of an input image $x$. Both are encoded by the shared encoder (the convolutional backbone network) $f$ into $y_1$ and $y_2$. Both augmentations are projected into $z_1$ and $z_2$ by a projection network $g$. A prediction network $h$ transforms $z_i$ into $p_i = h(z_i)$ and the whole network learns to match the output of the prediction network fed with the projection of one view $p_i = h(z_i)$ to the projection of the other view $z_j = g(y_j)$ and vice versa, by minimizing their negative cosine similarity:

$$\mathcal{D}(p_i, z_j) = -\frac{p_i}{\|p_i\|_2} \cdot \frac{z_j}{\|z_j\|_2}. \tag{1}$$

Most current SSL algorithms use a projection head $g$ on top of a convolutional feature extractor, as this was empirically found to help learn a much better final representation (Chen et al., 2020a;b; Grill et al., 2020b; Chen & He, 2020). However, it was also found that the learned projection is much worse in downstream tasks (Chen et al., 2020a) including for OOD (Sohn et al., 2020). Therefore, the output of the encoder backbone $f$ used during feature evaluation in most OOD methods is based on SSL, as it learns a much better representation. We call the outputs of $f$ the learned embedding.

### 2.2 IS REPRESENTATION QUALITY THE ONLY IMPORTANT THING FOR OOD ?

To illustrate that the quality of learned representation is only one factor for OOD performance, we train SimCLR on one class of CIFAR10 and evaluate the learned representation for OOD detection against all other classes; a similar setup is used in (Sehwag et al., 2021; Tack et al., 2020), however, we train for much longer. We repeat this experiment for 4 different classes, and evaluate the learned representations for OOD detection for each throughout training. Figure 2a (blue curve) shows the mean performance evaluation of the 4 classes; it is evident that performance reaches a peak very fast (within few hundred epochs) then deteriorates significantly as training progresses. Appendix A shows the same phenomena using 2 other evaluation metrics, with detailed graphs for each class.

One easy way to explain this behavior is the deteriorating quality of the learned representation. We carry out two more evaluations to examine this hypothesis. In Figure 2c we use linear evaluation (Zhang et al., 2016) and weighed k-NN (Wu et al., 2018) to evaluate the quality of learned representation. Figure 3d shows that the quality of the representation barely changes during training, i.e.

there is only the expected small decrease after the peak and then just fluctuations, and that linear separability holds. Weighted k-NN results supporting the same fact are in Appendix A.

These results tell two stories. (a) Methods based on density estimation show that performance drops down significantly as training progresses. (b) Doing supervised classification on the frozen features shows they are still of high quality and maintain nearly the same linear separable performance of the learned representation. This points to a change of the underlying distribution of the embedding space.

To study what is happening, we use the von Mises-Fisher (vMF) distribution which is a fundamental probability distribution on the $(n-1)$-dimensional hyper-sphere $S^{d-1} \subset R^d$. Its probability density function is $f_n(z, \mu, \kappa) = C_n(\kappa)e^{\kappa \mu^T z}$, where $\mu$ is the mean direction, and $\kappa$ is the concentration parameter. The vMF shape depends on $\kappa$: for high values, the distribution has a mode at the mean direction $\mu$; for $\kappa = 0$ it is uniform on the hyper-sphere $S^{d-1}$. vMF has been successfully used in both analyzing (Wang & Isola, 2020; Lee et al., 2018a; 2021) and learning (Hasnat et al., 2017; Davidson et al., 2018; Kumar & Tsvetkov, 2018) deep neural networks.

We fit a vMF (Banerjee et al., 2005; Sra, 2012) to the learned normalized embeddings of SimCLR, and study how $\kappa$ changes during training. Figure 2b shows how SimCLR starts with a relatively large $\kappa$ (high concentration) then reduces monotonically (low concentration, more uniform) as training progresses – this is true for both ID and OOD. In Figure 2f we notice the same behaviour using another tool: the Maximum Mean Discrepancy (MMD) (Gretton et al., 2012) between the learned representation and samples from a uniform distribution on a unit hypersphere, as suggested in (Sohn et al., 2020). MMD measures the distance between two probability distributions, so a high MMD means a less uniform and more concentrated distribution, and a low MMD the opposite.

Observe that this gives an explanation of the decaying performance seen in Figure 2a (blue curve). It is not the quality of the representation that is essential, but rather its distribution. As the ID data distribution gets more uniform, the probability $p$ to find an inlier sample $x \in$ ID arbitrarily close to an outlier query sample $x' \in$ OOD increases, i.e. ID and OOD get more and more indistinguishable. This is exactly the intuition behind many methods for AD, for example (Ruff et al., 2018; Goyal et al., 2020; Ruff et al., 2021)

Given this analysis, and the fact that contrastive methods are proven to converge to a uniform distribution (Wang & Isola, 2020), we take the natural step and perform the same analysis on a non-contrastive SSL method, SimSiam (selected over BYOL as it is simpler and retains most of the performance). Results are shown in Figure 2c, 2d, where indeed there is big difference from SimCLR: OOD performance stays nearly constant after reaching the peak. Also in Figure 2e, 2f we see that although the representation is also becoming more uniform, it is at a much slower rate, and is able to maintain high values for $\kappa$ and MMD even after many epochs of training.

It is also important to mention that this analysis shows that the number of training epochs (or use of early stopping criteria) is a very important hyper-parameter to choose for contrastive methods used for OOD. This needs to be carefully tuned, and we see that non-contrastive methods have a big advantage in being less sensitive to the number of epochs. Other works that noticed importance of early stopping in this context include (Reiss et al., 2020; Han et al., 2021; Reiss & Hoshen, 2021)

### 2.3 Learning a dense representation

(Schmidt et al., 2018) show that the sample complexity of robust learning can be significantly larger than that of standard learning. While some works tried to address this difference with extra positive or negative data, (Pang et al., 2019) propose the interesting idea of manipulating the local sample distribution of the training data via appropriate training objectives such that by inducing high-density feature regions, there would be locally sufficient samples to train robust classifiers and return reliable predictions. We propose pursuing the same direction with the OOD detection problem, and propose an architectural modification that can help induce high density feature space.

We propose adding a differentiable $l_2$-normalization operation after the encoder $f$ and before the projection head $g$. As such, the output $y_i = f(x_i)$ is transformed into $\hat{y}_i = y_i/||y_i||_2$ (as in Figure 1.2). The intuition is that adding the normalization step would deprive the model of any gradients when the norm of the embedding is changed, thus enforcing the model to learn directional transformations as those would be the only way to actually decrease the loss function. Regularizing the model this

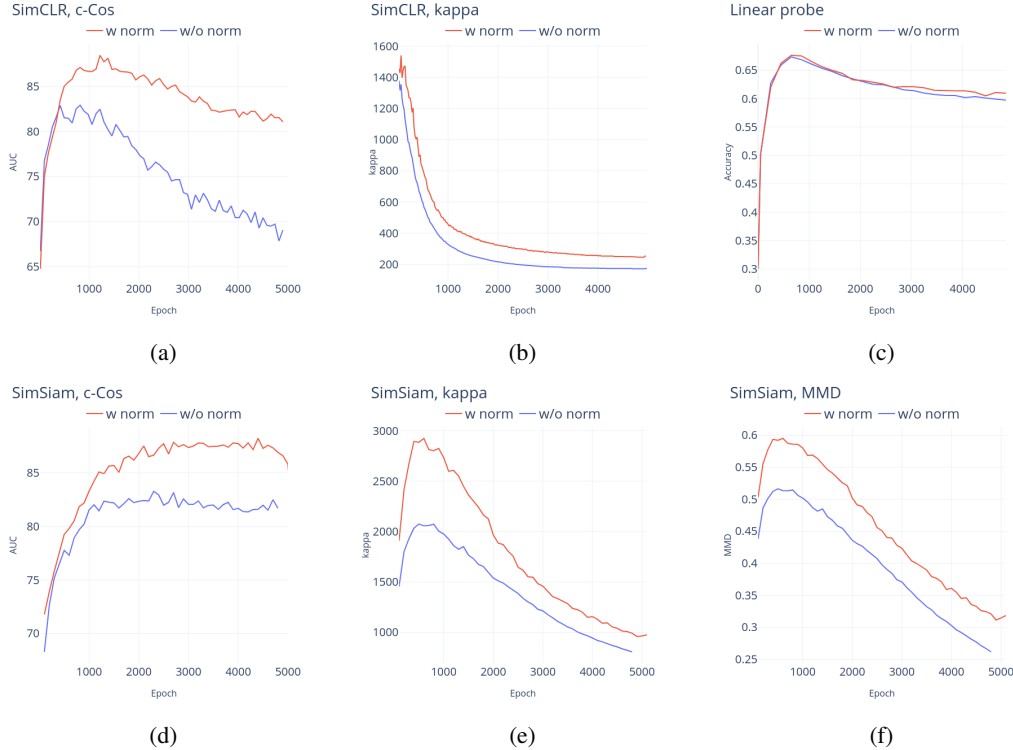

Figure 2: Analysis of training and evaluation of SimCLR and SimSiam on CIFAR10 for OOD with and without the proposed normalization. Every point on these plots represents the average from 4 independent classes. First row, SimCLR evaluated with (a) *c-Cos* (b) $\kappa$ from fitting a vMF (c) linear probe accuracy. Second row, SimSiam evaluated with (d) *c-Cos* (e) $\kappa$ from fitting a vMF (f) Maximum Mean Discrepancy (MMD) with a uniform distribution.

way should give a finer directional control on the learned embedding (the cosine distance makes more sense), and yield a more efficient use of volume and thus a denser representation.

We perform a series of experiments to empirically verify our intuition. Figure 2a (red curve) shows evaluation logs for SimCLR after adding normalization. It is evident the training is much more regularized now, the decrease in OOD performance after the peak is much smaller and performance is maintained for thousands of epochs. Figure 2b (red curve) shows a denser learned representation when compared to original SimCLR. We can see the same behaviour for SimSiam in Figure 2d, 2e.

One thing to note here that greatly strengthens the analysis made in the previous section, is the linear probe accuracy with and w/o norm (Figure 2c), they are essentially the same, even after 5000 epochs, on the other hand there is big deference between their OOD performance at that point (see Figure 2a). Further showing that it is a problem of feature distribution not feature quality.

Lastly, we would like to emphasize that while having a dense ID representation can be important for OOD detection with clean ID data, it is much more important in the presence of pollution in the ID data. In this polluted setup, some OOD data are mixed in during training and considered ID by the loss. In this case, a compact ID data distribution decreases chance of other OOD data to be considered as ID as much as possible. This is empirically verified in the experimental section.

## 2.4 SIMSIAM AND BYOL GRADIENT FLOW PROBLEM

To avoid the degenerate solution of a collapsed representation, the authors of SimSiam (Chen & He, 2020) found it crucial to have a gradient blocking operation (`stop-grad`) that blocks gradients starting from $\hat{z}_i$ from flowing back to other parts of the network. This lack of gradients acts as a regularizer and makes it hard for the optimizer to reach the trivial solution of a collapsed representation.

Note that the same analysis presented here equally applies to BYOL (Grill et al., 2020b): the existence of the momentum encoder enforces an implicit `stop-grad`.

However both (Chen & He, 2020; Grill et al., 2020b) do not study the effect `stop-grad` may have on proper gradient flow in the network. Studying Figure 1.2, we can see that the $l_2$-norm of the output of the prediction network $\hat{p_i} = p_i/||p_i||_2$ gives proper gradients that flows back to other parts of the network. The exact opposite happens for the $l_2$-norm of the encoder network's output: all gradients of that operation gets blocked by the `stop-grad`, though it is an integral part of the loss function.

**Proposed solution** Simply removing the `stop-grad` operation converges quickly to a collapsed representation (Chen & He, 2020). Another apparent straightforward solution is to minimize the norm of the output representation of the encoder $f$, but this limits the representation ability of the encoder and also rapidly collapses. One last trial would be removing normalization altogether in $\mathcal{D}$, however this converges to a sub-optimal representation with an unbounded norm (Grill et al., 2020b).

Our proposed solution is instead based on a simple observation. The missing gradient from $\hat{z_i}$ carries two pieces of information: (a) moving the output of the encoder $z_i$ to be closer to the output of the predictor $p_i$ (which is the information we want to hide from the optimizer). (b) encouraging the projector $g$ to learn a $l_2$-norm invariant representation; this facilitates training the predictor network and thus also the encoder, and is what we want to maintain.

In order to maintain (b) while discarding (a), we propose a small modification (see Figure 1.2): we apply a differentiable $l_2$-normalization to the projection $z_i$, such that the new projection is $\hat{z_i} = z_i/||z_i||_2$. This gives the network proper gradients for learning an $l_2$-normalized representation satisfying the loss function $\mathcal{D}$ while still having the `stop-grad` operation and avoiding related collapse.

## 2.5 FEATURE EVALUATION FOR OUT-OF-DISTRIBUTION DETECTION

Many recent works on OOD detection are based on scoring a given test sample using its distance to the nearest training sample (Tack et al., 2020; Sehwag et al., 2021; Lee et al., 2018b). This provides a simple evaluation baseline with strong results. Many metrics have been used for this, the most commonly used are the Mahalanobis distance (Lee et al., 2018b; Sehwag et al., 2021) and the Cosine distance (Tack et al., 2020) evaluated at the output of the network after the last convolutional block.

To take the advantages of both of these distances, we go a step further and score features with the arccosine of the Mahalanobis Cosine similarity (i.e. angular Mahalanobis distance) (Beveridge et al., 2005), which have been proposed and studied well previously in the field of face recognition (Beveridge et al., 2005; Vinay et al., 2015). Mahalanobis Cosine is the Cosine similarity between vectors after projection into the Mahalanobis space, where the famous Mahalanobis distance is the Euclidean distance computed between vectors after also projection into the Mahalanobis space.

If $x_m, y$ are a training and test sample respectively, and the projection of their features $f(x_m), f(y)$ into the Mahalanobis space is $u, v$ using the sample covariance matrix $\Sigma_m$ and mean $\mu_m$ of the training data $\{x_m\}_{m=1}^M$, then the Mahalanobis Cosine distance $\mathcal{D}_{MC}$ and our scoring $\mathcal{S}_{k\text{-}Cos}$ are

$$u = \Sigma_m^{-1/2}(x_m - \mu_m), \quad \mathcal{D}_{MC}(x_m, y) = \frac{u}{||u||_2} \cdot \frac{v}{||v||_2}, \quad \mathcal{S}_{k\text{-}Cos}(y) = arccos(\max_m \mathcal{D}_{MC}(x_m, y))$$

(2)

$\mathcal{S}_{k\text{-}Cos}$ considers only the distance to the nearest training sample and is used for most of our experiments. In Appendices C and G we study a variant $S_{c\text{-}Cos}$, that is especially robust to pollution; it computes distance to $\mu_m$, the mean vector of the training data $\{x_m\}$. $\mathcal{S}_{c\text{-}Cos}(y) = arccos(\mathcal{D}_{MC}(\mu_m, y))$.

**Feature ensembling** We also propose another evaluation scheme where all the intermediate feature maps of the network are scored independently, then their scores are all summed together. The idea is to get both high level (from final layers) and low level (from initial layers) OOD scores using different feature maps. Note that this is different from (Lee et al., 2018b) that learn a weighted sum of all the feature scores, where the weights are learned on a validation set; our proposed method is a simple sum and doesn't need a validation set. It also attains a huge runtime saving when compared to test-time augmentation (TTA) used in (Tack et al., 2020) or model ensembling used in (Sohn et al., 2020), as it requires a single model forward pass on a single instance. It consists of three steps:

1. Computing a score for each feature map, using either $\mathcal{S}_{k\text{-}Cos}$ or $\mathcal{S}_{c\text{-}Cos}$ or both.
2. Normalizing the range of the scores to be between 0 and 1, using the range of training data scores.
3. Summing the normalized scores.

Due to space limitations we show detailed feature evaluation results in Appendix C, along with comparing different evaluation metrics. We then show extensive ablations in Appendix G. In Section 3.2, we firstly consider experimental results without ensembling, before comparing to methods that use ensembling in Section 3.2.4. Any result that includes ensembling strictly has the $_E$ suffix, which indicates using the *Ens.* combination whose components are precisely described in Appendix C.

## 3 EMPIRICAL EVALUATION

### 3.1 EXPERIMENTAL SETUP

We perform a thorough evaluation of our proposed normalization modifications on SimSiam, BYOL, and SimCLR (with and without negative / shifted augmentations). The problem discussed in Section 2.4 doesn't apply to SimCLR (there is no stop-grad), so while both modifications are applied to SimSiam, our normalized variant of SimCLR only includes the change proposed in Section 2.3 for $\hat{y}$.

We evaluate in the one-vs-all OOD detection setting for CIFAR-10, CIFAR-100 super-classes (Krizhevsky et al., 2009), Fashion-MNIST (Xiao et al., 2017), and SVHN (Netzer et al., 2011). In this setting one class is treated as the normal class and the rest are treated as outliers.

For all results presented we use a ResNet-18, trained with Adam (Kingma & Ba, 2014) with a learning rate of 0.0001 and a cosine learning rate decay (Loshchilov & Hutter, 2016). For SimCLR we used 2-layer MLP as a projection head with an architecure similar to (Sohn et al., 2020). For SimSiam we used a 3-layer MLP for both the projector and the predictor. SimCLR is trained for 500 epochs, and SimSiam and BYOL for 4000 epochs. All models are trained on Nvidia V100 16GB GPUs and written in Pytorch (Paszke et al., 2019). All our reported results are without test-time augmentation.

For brevity, we often refer to SimSiam as **SS** and SimCLR as **SC**. Also **SS(n)**, **SC(n)** and **BYOL(n)** indicate inclusion of the proposed normalization, and **SC(-)** is with negative augmentations. Unless otherwise stated, our ensemble-free results use k-Cos Scoring. More details of the datasets, and additional comparisons/descriptions of competing methods can be found in Appendices B and F. An extensive ablation study including training 640 different models is provided in Appendix G.

### 3.2 RESULTS

#### 3.2.1 ONE-CLASS CLASSIFICATION

Table 1 compares ensemble-free versions of our baselines BYOL, SimSiam, and SimCLR with and without normalization against current state of the art (without ensembling) DROC (Sohn et al., 2020), RotNet (Golan & El-Yaniv, 2018), and GOAD (Bergman & Hoshen, 2020). Extended comparison of our results (without ensembling) with many other published methods can be found in Appendix F.

We see that adding normalization is consistently effective across BYOL, SS, and SC in all scenarios. Also, SS(n) and BYOL(n) always achieve very competitive results compared to methods that use negative augmentations (SC(-) and DROC) while making much less assumptions about the underlying training data.

#### 3.2.2 PERFORMANCE UNDER POLLUTION

Table 1 also shows a comparison of the performance under the very realistic scenario that some $p\%$ of the training data are polluted with OOD data. It can be seen that adding the proposed normalization drastically reduces the effect of polluted data, and achieves state-of-the-art results (without ensembling).

One important take-away from the table is that, as predicted by our analysis in Section 2.3, positive only SSL (e.g. SS or BYOL) is much more suited to real world OOD detection which can possibly

| Data | p | BYOL | BYOL(n) | SS | SS(n) | SC | SC(n) | SC(-) | SC(n-) | RotNet | DROC | GOAD |
|------|---|------|---------|-----|-------|-----|-------|-------|--------|--------|------|------|
| C10 | 0 | 88.5 | 90.5 | 89.5 | 91.7 | 86.3 | 88.9 | 90.7 | **92.9** | 89.3 | 92.5 | 88.2 |
| C10 | 0.1 | 82.9 | 85.3 | 83.2 | **86.3** | 65.6 | 79.8 | 79.6 | 83.9 | 78.5 | 80.5 | 83.0 |
| C100 | 0 | 79.6 | 80.2 | 81.4 | 84.3 | 80.5 | 84.0 | 84.7 | **87.0** | 81.9 | 86.5 | 74.5 |
| C100 | 0.1 | 76.2 | 77.8 | 78.9 | 80.3 | 75.9 | 80.0 | 80.5 | **82.8** | - | - | - |
| fMNIST | 0 | 95.3 | 95.1 | **95.9** | 95.0 | 94.6 | 94.9 | 94.7 | 95.7 | 94.6 | 94.5 | 94.1 |
| fMNIST | 0.1 | 61.3 | 73.2 | 63.0 | 75.3 | 46.5 | 53.1 | 78.7 | **80.9** | - | 76.6 | - |

Table 1: Results of baselines and proposed variants compared to state of the art, without ensembling. **p** is the ratio of outlier pollution data inside the training set. **SS** is SimSiam, **SC** is SimCLR.

| | | $BYOL(n)_E$ | $SS(n)_E$ | $SC(n)_E$ | $SC(n\text{-})_E$ | CSI | STOC |
|---|---|---|---|---|---|---|---|
| # Inference steps / example | | 1 | 1 | 1 | 1 | 160 | 1 |
| # Trained models / class | | 1 | 1 | 1 | 1 | 1 | 60 |
| Requires a good approximation of p | | ✗ | ✗ | ✗ | ✗ | ✗ | ✓ |
| Assumes rotation-variant data | | ✗ | ✗ | ✗ | ✓ | ✓ | ✓ |
| *Data* | *p* | | | | | | |
| CIFAR10 | 0.0 | 91.9 | 92.5 | 90.3 | 93.0 | 94.3 | 92.1 |
| CIFAR10 | 0.1 | 88.3 | 88.5 | 86.7 | 87.8 | 84.5 | 89.9 |
| fMNIST | 0.0 | 96.2 | 96.1 | 96.3 | 95.9 | - | 95.5 |
| fMNIST | 0.1 | 87.9 | 87.8 | 87.5 | 90.9 | - | 85.7 |
| CIFAR100 | 0.0 | 83.4 | 86.6 | | 89.4 | 89.6 | - |
| CIFAR100 | 0.1 | 80.7 | 82.5 | 83.0 | 85.7 | - | - |

Table 2: Performance compared to state of the art under different ensembling setups. p is the ratio of outliers inside the training data.

contain a small subset of polluted data compared to contrastive SSL (without negative augmentations) which suffers big performance drops in this scenario. In the case where negative augmentations are a suitable assumption, proposed SC(n-) gets even better results.

### 3.2.3 EFFECT OF NORMALIZATION

Table 3 examines the effect of the two proposed normalizations on SS on different batch size to asses the stability of training. As we can see for both batch-sizes both proposed normalizations do help the performance, and their combinations (proposed) is the best. We can also see for low batch sizes SS without norm suffers a big performance loss, which the proposed normalization fixes; this is a very important property as large batch training is not always an option.

### 3.2.4 EFFECT OF FEATURE ENSEMBLING

Table 2 shows results of the feature ensembling proposed in Section 2.5, compared to state-of-the-art techniques that also utilize ensembling: CSI (Tack et al., 2020) and STOC (Yoon et al., 2021). For

| | CIFAR10 | | CIFAR100 | |
|---|---|---|---|---|
| batch size | 32 | 512 | 32 | 512 |
| without norm | 77.97 | 89.56 | 74.46 | 81.38 |
| only normalize $f$ | 85.76 | 91.63 | 78.7 | 82.27 |
| only normalize $g$ | 81.73 | 89.22 | 73.1 | 81.15 |
| normalzie both | 92.9 | 91.7 | 81.73 | 84.31 |

Table 3: An ablation of SimSiam showing the importance of both the normalization schemes proposed in Section 2.3 and 2.4, especially for in low batch-size training.

| Trained from scratch | | Pre-trained (1M images) | | | | Pre-trained (1B images) | Pre-trained and adapted |
|---|---|---|---|---|---|---|---|
| | | Supervised | | | Self-supervised | | |
| SS(n) | SC(n) | PT R18 | PT R50 | PT R152 | PT R50 (SC) | PT R50 | PT R50 |
| 94.9 | 93.8 | 61 | 66 | 64 | 65 | 70 | 68 |

Table 4: OOD detection performance of various pre-trained backbones on SVHN

fair comparison, we also state the relative computational budget (training or inference) for each. The proposed feature ensembles offer big computational savings compared to CSI and STOC at roughly the same scores. Lastly, improvements by ensembling are more significant when there are outliers in the training data.

### 3.2.5 PRETRAINED BACKBONES ON SVHN

Some concurrent works (Reiss & Hoshen, 2021; Fort et al., 2021) claim that ImageNet pre-trained models can act as a universal OOD detector and can work well on nearly any in-domain distribution. (Anonymous, 2022) showed that for classes not present in the pre-training data, pre-trained models perform poorly at OOD detection.

We confirm this here, and show that even for very simple datasets (e.g. SVHN) that require different kinds of discriminative features than those required for natural images, pre-trained models perform very poorly. Table 4 shows that regardless of the model size (ResNet 18 to 152) or size of pre-training dataset (from 1M images to 1B images (Yalniz et al., 2019)), or using fully-supervised or SSL pre-training, or even with state-of-the-art feature adaptation after pre-training (Reiss & Hoshen, 2021), all catastrophically fail at SVHN compared to training from scratch, which can get a nearly perfect score (we report other related pre-training results from our implementation in Appendix F, where as expected performance is good in situations the OOD task is similar enough to the pretraining dataset; we see this as not a truly practical OOD detection benchmark however).

## 4 CONCLUSION

We have considered a general framework to detection of Anomalies in images: using various SSL methods as deep feature extractors; followed by metric learning for outlier scoring. For each stage we have studied what does and doesn't work in a variety of scenarios, and proposed remedies and improvements that are also robust in the case of polluted training data and small batch sizes.

We have investigated and studied compactness of ID representation distributions as an important and very sensitive factor to the final OOD detection performance. Our experiments demonstrated that regardless of the quality of learned features, the ID representation compactness is critical. As its distribution gets closer to uniform, the OOD detection performance deteriorates significantly.

We have motivated, proposed, and studied an assumption-free, novel architectural modification for inducing this non-uniformity, and use it to solidly improve performance across contrastive and non-contrastive SSL-based OOD detection. We also studied several variants of feature scoring that work well across these different methods. More importantly, under the real world setting of totally unsupervised AD, where the ID training data can be polluted by some OOD outliers, our proposed modifications provide state-of-the-art performance among all competing methods.

Previous state-of-the-art literature for OOD detection was based on contrastive-based SSL with negative "distributionally-shifted" augmentations (e.g. 90 degree rotations). A big hurdle with the applicability of these methods is that the assumptions they make about the training data can be partially or fully invalid in real-world scenarios. Using our proposed "No Shifted Augmentations" (NSA) modifications, both contrastive and non-contrastive methods get a boost in their baseline performance, making them comparable to negative augmentation based techniques, but much more applicable to open world scenarios where little is controlled about ID data.

While model ensembling or Test-Time Augmentation is known in literature to be very effective for OOD, increased training/inference computational requirements can be often prohibitive. We went further and studied using light-weight multi-level feature ensembling for OOD. This enabled us to show state-of-the-art performance in terms of AUCROC, with huge savings in computational budget.

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

## A    PLOTS FOR EVALUATIONS DURING TRAINING

Here we show the detailed version of the summary (mean) plots presented in Figure 2 in the main text. They show the same curves for 4 distinct classes of CIFAR10, and also under 3 different metrics. They appear in Figure 3 (SimCLR AUC and Linear probes during training), Figure 4 (SimCLR Kappa and MMD during training), Figure 5 (SimSiam AUC during training), Figure 6 (SimSiam Kappa and MMD during training), and Figure 7 (SimCLR with norm AUC). The same findings as in the main text are seen to hold across these detailed variants.

## B    DATASET DETAILS

The main datasets we test on are: CIFAR 10, CIFAR 100 and ImageNet30, fashion-MNIST following the splits and protocols specified in (Tack et al., 2020) and (Sohn et al., 2020); we add SVHN using the same basic protocol of one class as inlier v.s. the remaining as outliers.We resized images to 32 x 32 for all datasets apart from ImageNet30 which uses the standard ImageNet ResNet-18 architecture's transformation of a 224-pixel center-cropped region from the 256 x 256 input image.

## C    COMPARISON OF SCORING WITH DIFFERENT FEATURE EVALUATIONS, METRICS & ENSEMBLING

As discussed in Section 2.5, we compare the default scoring used in our main experiments (the encoder's last layer features using $\mathcal{S}_{k\text{-}Cos}$) with a feature ensembling evaluation scheme consisting of these scores summed across feature maps from the encoder backbone network, and projection heads from either SimSiam or SimCLR. In the following tables we use following feature maps:

- conv_block_$n$: The output feature map from block$n$ of the convolutional backbone. Which is first directly flattened from 2D to 1D before evaluation.

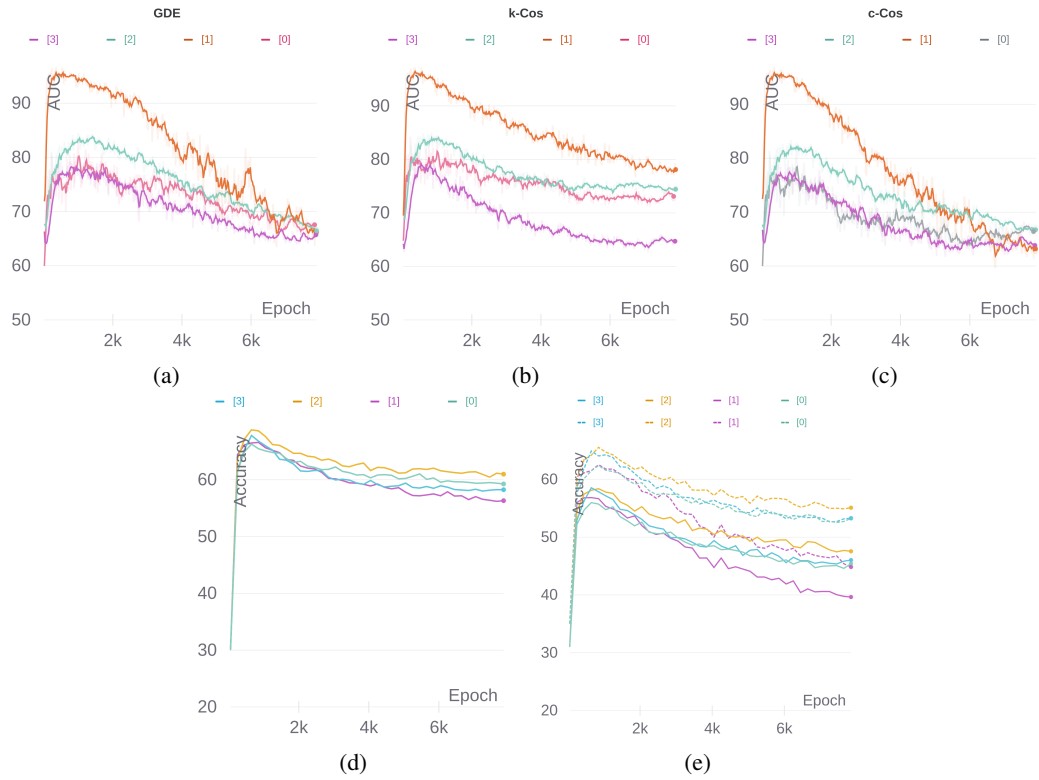

Figure 3: Training SimCLR on classes 0,1,2,3 from CIFAR10 and evaluating the representation during training. (a) Using Gaussian density estimation (GDE) (Reynolds, 2009). (b) $k$-Cos cosine distance to the closest ($k$=1) point in ID data. (c) $c$-Cos cosine distance to the mean direction in ID data. Both these are defined in more detail in Section 2.5. (d) Linear evaluation (Zhang et al., 2016) of learned embedding. (d) Weighted nearest neighbor classifier (k-NN) (Wu et al., 2018). Solid lines represent $k = 1$ and dashed lines represent $k = 20$

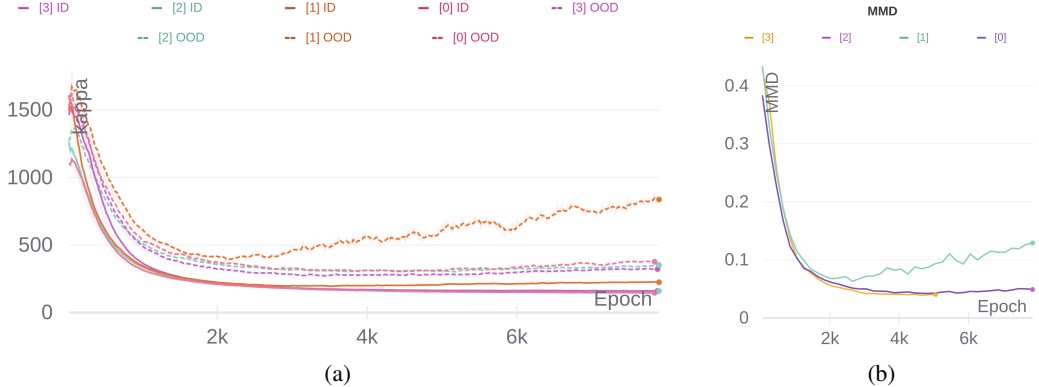

Figure 4: Training SimCLR on classes 0,1,2,3 from CIFAR10 and monitoring the learned vMF ID and OOD representations during training. (a) $\kappa$ progress during training; solid lines are ID data and dashed are OOD data. (b) The MMD (Gretton et al., 2012) between the learned representation and samples from uniform distribution on a unit hypersphere (lower is more uniform).

- conv_block_$n$ (1x1): Same as conv_block_$n$ but after pooling to a size of 1x1.
- head_layer_$n$: The output feature map from layer $n$ of the projection head.
- All Conv blocks: The sum of the scores of all convolutional blocks, using the distance specified in column, then summed across table columns merged in table.

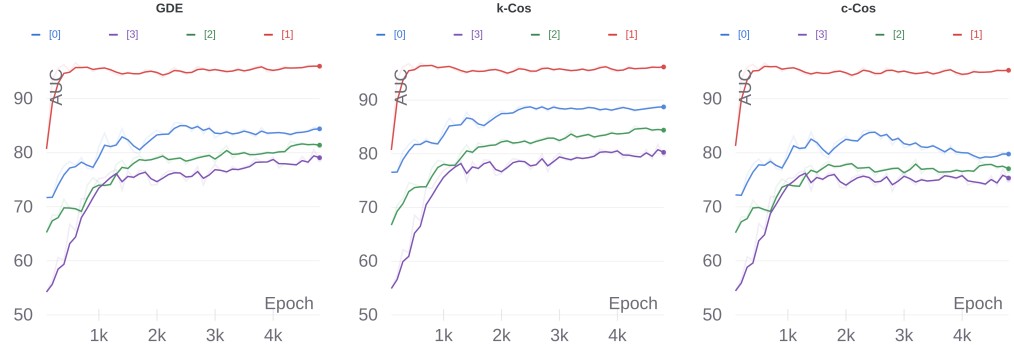

Figure 5: Same experiment as Figure 3 but with SimSiam.

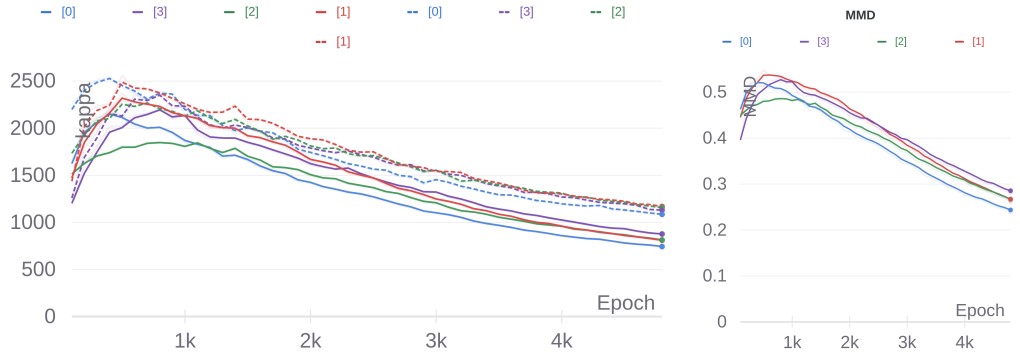

Figure 6: Same experiment as Figure 4 but with SimSiam.

- All blocks: The sum of the scores of all network internal feature maps, using the distance specified in column, then summed across table columns merged in table.
- Ens.: The sum of *k-Cos* and *k-Cos (Mah)* for 2D feature maps (e.g. convolutional feature maps) and *c-Cos* and *c-Cos (Mah)* for 1D feature maps (e.g. projection heads).

We also investigate 5 different metric functions for computing the OOD score, namely:

- *k-Cos*: Cosine distance to closest (k=1) training vector, introduced in Section 2.5
- *k-Cos (Mah)*: Same but evaluated in Mahalanobis space.
- *c-Cos*: Cosine distance to the mean of all the training set vectors, introduced in Section 2.5.
- *c-Cos (Mah)*: Same but evaluated in Mahalanobis space.
- *GDE*: A Gaussian Kernel density estimator, we use the same configuration as in Sohn et al. (2020).

Tables 6,7,8 show an extensive evaluation of presented models at most internal layers and a variety of scoring metrics. We should highlight the fact that each number presented in those tables is the average across all classes in the dataset, with one experiment per class i.e. 10 experiments for CIFAR10, and 20 experiments for CIFAR100. We can notice:

- The Cosine Mahalanobis distance mostly outperforms most other evaluation metrics, sometimes a by large gap.
- There is a consistent improvement associated with ensembling of feature scores. This is true across models and across datasets.
- We note this type of feature ensembling can be considered free, computation-wise, compared with the ensembling used in e.g. CSI, that slows down the method significantly at inference.

| | | CIFAR10 | CIFAR100 | IN30 | fMNIST | SVHN |
|---|---|---|---|---|---|---|
| **Ours** | 1. **SS(n)** = SimSiam with norm, aka "NSA" | 91.54 | 84.09 | 80.88 | 95.15 | 94.91 |
| | 2. **SC(n)** = SimCLR(w norm, no neg aug) | 89.27 | 83.95 | 75.17 | 94.61 | 93.84 |
| | 3. **SC)** = SimCLR(w/o norm, no neg aug) | 86.49 | 80.51 | 75.17 | 94.61 | 93.84 |
| | 4. **SS(-)** = SimCLR(w/o norm, neg aug) | 91.12 | 86.68 | 76.97 | 95.47 | 92.17 |
| **Pretraining / FT** | 5. Pretrained ResNet18* [Ours] | 93.63 | 92.64 | 99.78 | 94.35 | 61.06 |
| | 6. Pretrained ResNet50* [Ours] | 94.45 | 94.68 | 99.91 | 95.16 | 66.28 |
| | 7. Pretrained ResNet152* [Ours] | 95.82 | 95.21 | 99.96 | 94.99 | 63.41 |
| | 8. PT ResNet50 (DROC (Sohn et al., 2020))* | 80.0 | 83.7 | - | 91.8 | - |
| | 10. ResNet152 (DN2 (Reiss et al., 2020))* | 92.5 | 94.1 | - | 94.5 | - |
| | 11. ResNet152 (PANDA (Reiss et al., 2020))* | 96.2 | 94.1 | - | 95.6 | - |
| | 12. Self-Sup. PT R50. Xiao et al.(Xiao et al., 2021)* | 93.8 | 92.6 | - | 94.4 | - |
| **Dist. Shifting** | 13. CSI (SimCLR loss only)** | 87.9 | - | - | - | - |
| | 13. CSI (SimCLR w neg aug only)** | 90.1 | 86.5 | 83.1 | - | - |
| | 14. CSI (Full)** | 94.3 | 89.6† | 91.6 | - | - |
| | 15. DROC, Contrastive | 89.0 | 82.4 | - | 93.9 | - |
| | 16. DROC, Contrastive DA | 92.5 | 86.5 | - | 94.8 | - |
| **Other Methods** | 19. DeepSVDD (Ruff et al., 2018) | 64.8 | 67.0 | - | 84.8 | - |
| | 20. DROCC (Goyal et al., 2020) | 74.2 | - | - | - | - |
| | 21. Geom (Golan et al.) (Golan & El-Yaniv, 2018)** | 86.0 | 78.7 | - | 93.5 | |
| | 22. GOAD (Bergman) (Bergman & Hoshen, 2020)** | 88.2 | 74.5 | - | 94.1 | - |
| | 23. ARNet (Huang) (Huang et al., 2019)** | 86.6 | 78.8 | - | 93.33 | |
| | 24. Hendryks et al. (Hendrycks et al., 2019)** | 90.1 | 79.8 | 85.7 | 93.2 | - |
| | 25.SSD (Sehwag et al., 2021) | 90.0 | - | - | - | - |

Table 5: One-Class Classification Summary results reported in the literature on various datasets, plus some of our results; all figures are AUC. * indicates methods trained on external additional data, which may overlap in scope with the "unseen" OOD data. ** indicates method using test-time data augmentation / ensembling during evaluation, which can involve drastically slower inference.

| | k-Cos | k-Cos (Mah) | c-Cos | c-Cos (Mah) | GDE |
|---|---|---|---|---|---|
| conv_block_2 | 80.29 | 83.58 | 65.35 | 83.52 | |
| conv_block_3 | 85.87 | 83.59 | 75.81 | 83.53 | |
| conv_block_4 | 92.21 | 91.20 | 88.35 | 91.12 | |
| conv_block_1 (1x1) | 71.29 | 70.85 | 68.39 | 67.37 | 67.39 |
| conv_block_2 (1x1) | 73.19 | 71.93 | 70.94 | 69.66 | 69.39 |
| conv_block_3 (1x1) | 81.46 | 83.47 | 73.60 | 79.66 | 79.32 |
| conv_block_4 (1x1) | 91.85 | 91.54 | 84.88 | 90.52 | 91.04 |
| head_layer_1 | 82.64 | 87.84 | 71.18 | 87.56 | |
| head_layer_2 | 80.62 | 83.08 | 70.50 | 82.76 | |
| head_layer_3 | 79.67 | 77.41 | 51.93 | 54.64 | |
| All Conv blocks | 92.37 | | 90.79 | | |
| All Conv blocks | 92.01 | | | | |
| All blocks | 91.62 | 92.23 | 80.50 | 90.86 | 89.96 |
| All blocks | 92.86 | | 88.74 | | |
| All blocks | 92.86 | | | | |
| Ens. | 92.5 | | | | |

Table 6: Different feature ensembling methods: SimSiam w norm on Cifar10. Note that items spanning multiple columns imply summation of the corresponding features.

|                     | k-Cos | k-Cos (Mah) | c-Cos | c-Cos (Mah) | GDE   |
|---------------------|-------|-------------|-------|-------------|-------|
| conv_block_2        | 73.36 | 76.42       | 58.54 | 76.39       |       |
| conv_block_3        | 79.31 | 77.83       | 68.39 | 77.80       |       |
| conv_block_4        | 85.47 | 84.46       | 78.19 | 84.40       |       |
| conv_block_1 (1x1)  | 66.35 | 67.51       | 60.75 | 64.77       | 64.76 |
| conv_block_2 (1x1)  | 68.05 | 67.99       | 61.93 | 65.46       | 65.37 |
| conv_block_3 (1x1)  | 74.26 | 76.49       | 64.76 | 72.78       | 72.75 |
| conv_block_4 (1x1)  | 84.76 | 84.09       | 72.08 | 82.74       | 83.15 |
| head_layer_1        | 76.95 | 81.46       | 64.50 | 81.24       |       |
| head_layer_2        | 74.78 | 77.46       | 62.95 | 77.29       |       |
| head_layer_3        | 75.23 | 73.86       | 52.65 | 68.15       |       |
| All Conv blocks     | 86.45 |             | 84.75 |             |       |
| All Conv blocks     | 85.89 |             |       |             |       |
| All blocks          | 85.19 | 87.11       | 72.50 | 86.17       |       |
| All blocks          | 87.54 |             | 85.79 |             |       |
| All blocks          | 87.25 |             |       |             |       |
| Ens.                | 86.6  |             |       |             |       |

Table 7: Different feature ensembling methods: SimSiam w norm on Cifar100

|                     | k-Cos | k-Cos (Mah) | c-Cos | c-Cos (Mah) | GDE   |
|---------------------|-------|-------------|-------|-------------|-------|
| conv_block_2        | 79.62 | 82.40       | 64.71 | 82.32       |       |
| conv_block_3        | 81.08 | 80.22       | 73.38 | 80.14       |       |
| conv_block_4        | 88.92 | 89.47       | 82.83 | 89.42       |       |
| conv_block_1 (1x1)  | 71.63 | 71.53       | 68.56 | 68.12       | 68.17 |
| conv_block_2 (1x1)  | 73.85 | 72.67       | 70.60 | 70.12       | 69.79 |
| conv_block_3 (1x1)  | 74.97 | 77.00       | 68.97 | 74.47       | 73.50 |
| conv_block_4 (1x1)  | 86.68 | 89.27       | 75.68 | 88.43       | 88.39 |
| head_layer_1        | 73.88 | 81.05       | 48.36 | 80.63       |       |
| head_layer_2        | 65.19 | 72.33       | 41.26 | 46.18       |       |
| All Conv blocks     | 90.65 |             | 89.61 |             |       |
| All Conv blocks     | 90.57 |             |       |             |       |
| All blocks          | 87.37 | 90.16       | 75.65 | 89.12       |       |
| All blocks          | 90.20 |             | 89.10 |             |       |
| All blocks          | 90.35 |             |       |             |       |
| Ens.                | 90.3  |             |       |             |       |

Table 8: Different feature ensembling methods: SimCLR w norm on Cifar10

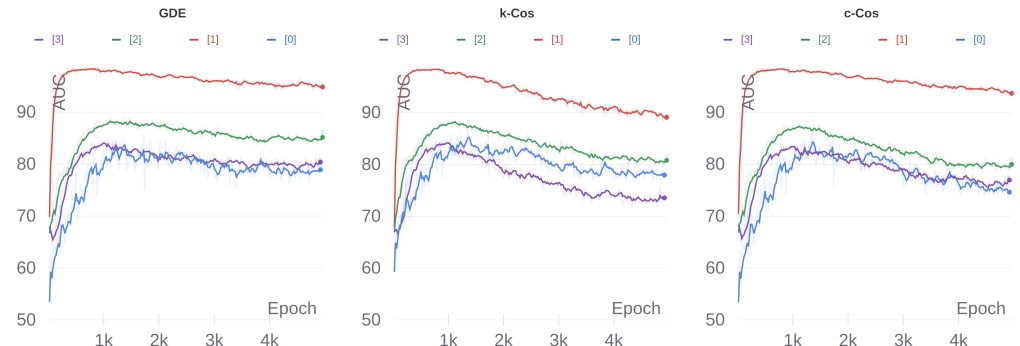

Figure 7: Same experiment as Figure 3 but after adding the normalization proposed in Section 2.3.

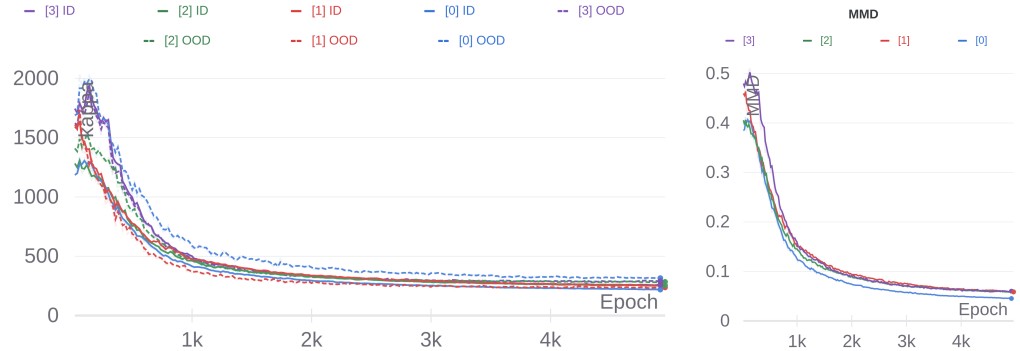

Figure 8: Same experiment as Figure 4 but after adding the normalization proposed in Section 2.3.

## D   CIFAR 10 PER-CLASS RESULTS

Here in Table 9 we show the per class results for each of the CIFAR 10 classes trained for one-class classification against the others, and compare our NSA method (SimSIAM with Norm, no ensembling augmentations), with (i) last layer features only and (ii) summing all features, with other recent works that also report these results.

## E   ADDITIONAL POLLUTION RESULTS

Table 10 shows additional CIFAR10 results in the presence of pollution, also more comparisons.

## F   ADDITIONAL COMPARISON WITH PREVIOUS RESULTS FROM LITERATURE

Table 5 gives a more detailed overview of related OOD / One Class Classification results from the literature, and comparison with variants of our baseline (ensemble free) variants of SimSiam, SimCLR,

| | Plane | Car | Bird | Cat | Deer | Dog | Frog | Horse | Ship | Truck | Mean |
|---|---|---|---|---|---|---|---|---|---|---|---|
| CSI (Tack et al., 2020), inc. ensembling augmentations | 90.0 | 99.1 | 93.2 | 86.4 | 93.8 | 93.4 | 95.2 | 98.6 | 97.9 | 95.5 | 94.3 |
| DROC, OC-SVM (Sohn et al., 2020) | 88.8 | 97.5 | 87.7 | 82.0 | 82.4 | 89.2 | 89.7 | 95.6 | 86.0 | 90.6 | 89.0 |
| DROC OC-SVM (DA) (Sohn et al., 2020) | 91.0 | 98.9 | 88.0 | 83.2 | 89.4 | 90.0 | 93.5 | 98.1 | 96.5 | 95.1 | 92.5 |
| DROC Gaussian KDE (DA) (Sohn et al., 2020) | 91.0 | 98.9 | 88.0 | 83.2 | 89.4 | 90.2 | 93.5 | 98.1 | 96.5 | 95.1 | 92.4 |
| Rot. Pred. OC-SVM, from (Sohn et al., 2020) | 83.6 | 96.9 | 87.9 | 79.0 | 90.5 | 89.5 | 94.1 | 96.7 | 95.0 | 94.9 | 90.8 |
| Denoising OC-SVM, from (Sohn et al., 2020) | 81.6 | 92.4 | 75.9 | 72.3 | 82.3 | 83.1 | 86.7 | 91.2 | 78.0 | 91.0 | 83.4 |
| RotNet Rot. Cls, (Golan & El-Yaniv, 2018) | 80.3 | 91.2 | 85.3 | 78.1 | 85.9 | 86.7 | 89.6 | 93.3 | 91.8 | 86.0 | 86.8 |
| NSA (SimSIAM w norm) [Ours] | 90.4 | 98.6 | 85.2 | 85.7 | 84.1 | 92.9 | 92.9 | 94.5 | 96.3 | 90.99 | 91.52 |
| NSA (SimSIAM w norm) All features [Ours] | 93.07 | 98.44 | 87.16 | 83.81 | 90.34 | 91.79 | 96.79 | 96.16 | 94.80 | 96.29 | 92.86 |

Table 9: CIFAR10 Per class results

|  | p=0 | p=0.05 | p=0.10 | $\Delta$ (p=0.1 - p=0) |
|---|---|---|---|---|
| SimSiam(w norm) | 91.54 | 88.08 | 86.21 | 5.33 |
| SimSiam(w/o norm) | 89.56 | 85.21 | 81.79 | 7.77 |
| SimCLR(no neg, w norm) | 89.27 | 83.1 | 77.49 | 11.78 |
| SimCLR(no neg, w/o norm) | 86.49 | 71.1 | 63.5 | 25.99 |
| SimCLR (neg aug, w norm) | 92.91 | 86.1 | 83.38 | 9.53 |
| SimCLR (neg aug, w/o norm) | 91.12 | 81.5 | 77.94 | 13.18 |
| Pretrained IN18 | 92.63 | 90.9 | 89.3 | 3.33 |
| Pretrained IN50 | 94.45 | 92.18 | 89.49 | 4.96 |
| Pretrained IN152 | 95.82 | 94.48 | 91.39 | 4.43 |
| DROC (Sohn et al., 2020) | 89 | 76.5 | 73.0 | 16.0 |
| DROC (DA) (Sohn et al., 2020) | 92.5 | 85.0 | 80.5 | 12.0 |
| CSI (results from (Han et al., 2021)) | 94.3 | 88.2 | 84.5 | 9.8 |
| Semi-Supervised (5% labeled) ELSA (Han et al.) | 85.7 | 83.5 | 81.6 | 4.1 |
| Semi-Supervised (5% labeled) ELSA+ (Han et al.) | 95.2 | 93.0 | 91.1 | 4.1 |

Table 10: CIFAR10 pollution experiments. p is the ratio of outlier data inside the training set. the last column is the loss in performance between training with clean data and a 10% polluted data. Clearly, our proposed modifications reduces the drop in all cases. With SimSiam we beat standard ELSA which uses 5% labeled data to maintain robustness to pollution, and come close to ELSA+, which uses TTA and other tricks from CSI on top of the baseline SSL approach.

along with our implementation of pretrained ResNets (as an additional basline for comparison to other pretrained methods.

The pretrained results also serve to indicate where these types of approaches can fall down, and the issues in a with fairly comparing these pretrained methods with from-scratch trained approaches in the Anomaly/OOD detection context (It is unsurprising that pretraining a representation on ImageNet does well at distinguishing "unseen" ImageNet30 classes in a One-Class classifier scenario, and similarly for e.g. CIFAR, where the same types of class are present; however SVHN is less similar and therefore the pretrained represenation is less useful).

More details and notes on these competing methods follow. In particular, we clarify that "**(n)**" indicates a method using our proposed modifications, and "w/o norm" is the standard version of the architecture (SimCLR or SimSiam) without these modifications. "'**(-)**" means including strong distributionally negative shifted augmentations, using randomly the four 0, 90, 180, and 270 degree rotations, following the approach of CSI (Tack et al., 2020).

We note in particular that CSI's method combines many parts (contrastive+classification losses and scoring functions, plus ensembling) which each contribute something and add up to give good results. Our goal of instead showing baseline SimCLR results with / without the norm, and with/without shifted augmentations is to create a more straightforward baseline to compare with. We note improved results are possible with our method adding these additional features such as ensembling, but also at additional computational cost, as is the case with CSI.

On the other hand, our pretrained baselines obtain very good results equal or exceeding PANDA (Reiss et al., 2020)'s fine-tuned results on datasets that are similar in nature to ImageNet that they are pretrained on; this exemplifies the benefits of our chosen scoring metric $\mathcal{S}_{NSA} = \mathcal{S}_{k\text{-}Cos}$ on good representations in general. However we also show datasets where this approach falls down, compared with self-supervised methods.

For each of the methods, here is a more detailed description of the features, networks, scoring functions etc. used for comparison:

1. NSA [ours]; features = SimSiam (with norm), ResNet18; scoring = KNN + Mahalanobis Cosine, last layer features.

2. features = SimCLR(w norm) + **NO** negative shifting augmentations, ResNet18 ; scoring = KNN + Mahalanobis Cosine, last layer features. [our baseline]

3. features = SimCLR(w/o norm) + **NO** negative shifting augmentations, ResNet18 ; scoring = KNN + Mahalanobis Cosine, last layer features. [our baseline]

4. features = SimCLR(w/o norm) + **With** strong rotation negative shifting augmentations, ResNet18; scoring = KNN + Mahalanobis Cosine, last layer features [our baseline, closest to CSI (**without ensembling**) / DROC+DA]

5. Pretrained ResNet18 (on ImageNet), **no fine-tuning**; scoring = KNN + Mahalanobis Cosine, last layer features [our baseline]

6. Pretrained ResNet50 (on ImageNet), **no fine-tuning**; scoring = KNN + Mahalanobis Cosine, last layer features [our baseline]

7. Pretrained ResNet152 (on ImageNet), **no fine-tuning**; scoring = KNN + Mahalanobis Cosine, last layer features [our baseline]

8. Pretrained ResNet-50 on ImageNet, results from DROC (Sohn et al., 2020)

9. Rippel et al. (Rippel et al., 2020) - Pretrained EfficientNet-B4 on Imagenet + Mahalanobis (no results on our benchmark datasets).

10. Reiss et al (Reiss et al., 2020) Simple baseline "DN2" - Pretrained ResNet-152 on ImageNet, **no fine-tuning** + kNN=2 scoring from last layer, presumably euclidean distance

11. Reiss et al. (Reiss et al., 2020) PANDA-EWC - Pretrained ResNet-152 on ImageNet, **Fine-tuned** last 2 layers on each dataset, with compactness loss + kNN=2 scoring from last layer, presumably euclidean distance.

12. Xiao et al.(Xiao et al., 2021) - Self-supervised Pretrained ResNet-50. SimCLRv2, Gaussian Mixture Model / Mahalanobis distance .

13. (a) CSI, SimCLR loss only; features = SimCLR (w/o norm), ResNet18 ; scoring = Sim-only contrastive(cosine * norm). [their results Table 15]; (b) same but with full CSI loss, contrastive **With** strong rotation negative shifting augmentations; scoring = contrastive Sim-only (cosine).

14. Full CSI; features = SimCLR(w/o norm) + **With** strong rotation negative shifting augmentations, ResNet18 ; scoring = contrastive(cosine * norm) + rotation prediction, on shifted transforms, **with ensembles**. [their main results, essentially combining many parts]

15. DROC (Sohn et al., 2020) Contrastive (=Deep Representation One-class Classification); features = SimCLR(w/o norm) + **NO** negative shifting augmentations ; scoring = (OC-SVM). [their results]

16. DROC (Sohn et al., 2020) Contrastive DA (=Deep Representation One-class Classification); features = SimCLR(w/o norm) + **With** strong rotation negative shifting augmentations ; scoring = (OC-SVM). [their results]

17. ELSA (Han et al., 2021) - **NO** negative shifting augmentations - with 1% labeled outliers (Han et al., 2021)

18. ELSA+ (Han et al., 2021) - **With** strong rotation negative shifting augmentations (like CSI), and **with ensembles** - also with 1% labeled outliers (Han et al., 2021)

19. DeepSVDD, Ruff et al. (Ruff et al., 2018), with LeNet architecture Autoencoder + adapted features. [All results apart from CIFAR10 from Reiss(Reiss et al., 2020)]

20. DROCC, Goyal et al. (Goyal et al., 2020) - LeNet architecture

21. Geom - Golan et al. (Golan & El-Yaniv, 2018) - WRN-16-8 Arch.

22. GOAD, Bergman et al. (Bergman & Hoshen, 2020) - WRN-16-4 architecture [CIFAR 100 results from CSI with ResNet18] (Bergman & Hoshen, 2020)

23. ARNet (formerly called Inv. Trans AE) - Huang et al Huang et al. (2019)

24. Rot + Trans, Hendryks et al. (Hendrycks et al., 2019) - WRN-16-4 architecture (CIFAR 100 results from CSI with ResNet18; IN30 are ResNet18 Rot+Trans+Attn+Resize; fMNIST results from Reiss (Reiss et al., 2020))

25. SSD (Sehwag et al., 2021); features = SimCLR(w/o norm) + **NO** negative shifting augmentations ; scoring = Mahalanobis [their results]

## G    DETAILED ABLATION STUDY

In Table 11 we show an extensive ablation study demonstrating results on CIFAR10, CIFAR100 and fMNIST of each variant of the methods we study, with and without our normalization enhancements, for different pollution settings, and under 5 different feature evaluations metrics (same as those in Appendix C) and the feature ensemble (Ens.) proposed in Section 2.5. We would like to stress that this study includes training 640 different models and evaluating each model using 6 different metrics.

We can take a few important notes:

- For all examined situations, the proposed normalization always brings a noticeable improvement. The highest improvement is for SimCLR in the presence of pollution; this is consistent with our analysis in the main text.
- For different datasets, different algorithms, and different pollution ratios, the proposed ensembling has the best performance most of the time, and the second best otherwise, which shows how generic our proposed ensembling scheme is.
- k-Cos (Mah) is the metric most often getting second best results among all evaluated, and as such was chosen for our simple baseline (ensemble-free comparison).
- Although in many times it is not the best or second best metric, c-Cos is the metric that gets the largest boost in performance after applying normalization, which is expected because as the ID representation gets more compact, the center is much more representative of the distribution.

## H    BACKGROUND AND RELATED WORK

### H.1    ANOMALY AND OOD DETECTION

Outlier detection is important in a variety of practical tasks, such as detecting problems in a production process, detecting security events, and acting upon novelties. The most general case is unsupervised novelty detection (or poisoned data): there are outliers in the training set, and we have no information about them. Then there are degrees of supervision, semi-supervised (a few outliers are labeled) and fully supervised (all outliers are labeled). Furthermore, in the sub-field of anomaly detection it is assumed that there are no outliers in the training data. A challenge in OOD detection is that the notion of in- and out-of-distribution is not well defined, and task-dependent. A good OOD method would generalize to different notions of out-of-distribution and datasets, e.g. with respect to color, style, perspective and content. Recent approaches in Anomaly/OOD detection can be categorized in four groups:

- **Density-based methods** are based on the assumption that models trained to fit the in-distribution data will be less confident on out-of-distribution data in terms of likelihood of the outputs. Using the likelihood as a detection score has been shown to be a weak metric (Nalisnick et al., 2018; Choi et al., 2018), and modifications such as entropy, energy (Du & Mordatch, 2019; Grathwohl et al., 2019) and WAIC (Choi et al., 2018) have been proposed.
- **One-class classifiers** are a classic approach for outlier detection and have been adopted to deep learning settings. They find a decision boundary that separates ID and OOD samples. A margin is introduced to allow generalization (Schölkopf et al., 2000; Ruff et al., 2018).
- **Reconstruction-based methods** model the ID training data by training an encoder and decoder network to reconstruct the in-distribution data. The reconstruction will generalize less for OOD data such that the reconstruction loss can be used a the detection metric. Auto-encoders (Zong et al., 2018; Pidhorskyi et al., 2018) and GANs (Schlegl et al., 2017; Deecke et al., 2018; Perera et al., 2019).
- **Self-supervised methods** leverage the representations learned from self-supervision, combined with different detection scores. The current state-of-the-art in OOD detection is CSI (Tack et al., 2020), using representations learned by SimCLR (Chen et al., 2020a) and the distance to the closest training point in latent space as a detection score. Other approaches train networks with predefined tasks such as permutations of image patches or rotations (Golan & El-Yaniv, 2018; Hendrycks et al., 2019; Bergman & Hoshen, 2020)].

Table 11: Detailed ablation study. Here **p** is the ratio of outlier data inside the training set. **Norm** is whether normalization is applied or not. **Ens.** is our proposed feature ensembling. Best results are in **bold**. Second best results are underlined.

| Data | Algo | p | Norm | k-Cos | k-Cos (MAH) | c-Cos | c-Cos (MAH) | GDE | Ens. |
|------|------|---|------|-------|-------------|-------|-------------|-----|------|
| C10 | BYOL | 0 | ✗ | 84.9 | 88.5 | 51.3 | 83.8 | 86.2 | **88.9** |
| | | 0 | ✓ | 89.9 | 90.5 | 79.5 | 89.0 | 89.5 | **91.9** |
| | | 0.1 | ✗ | 62.7 | 82.9 | 63.4 | 82.0 | 80.5 | **85.3** |
| | | 0.1 | ✓ | 77.5 | 85.3 | 78.5 | 83.6 | 83.0 | **88.3** |
| | SimSiam | 0 | ✗ | 86.5 | 89.5 | 59.3 | 85.8 | 87.9 | **90.1** |
| | | 0 | ✓ | 91.6 | 91.7 | 84.9 | 90.5 | 91.0 | **92.5** |
| | | 0.1 | ✗ | 65.8 | 83.2 | 67.9 | 81.5 | 80.7 | **84.3** |
| | | 0.1 | ✓ | 80.6 | 86.3 | 82.5 | 84.9 | 84.3 | **88.4** |
| | SimCLR | 0 | ✗ | 79.3 | 86.3 | 45.6 | 84.9 | 84.7 | **87.8** |
| | | 0 | ✓ | 86.0 | 88.9 | 75.2 | 87.9 | 88.0 | **90.3** |
| | | 0.1 | ✗ | 53.1 | 65.6 | 62.6 | 65.9 | 64.7 | **80.3** |
| | | 0.1 | ✓ | 68.8 | 79.8 | 80.0 | 78.7 | 78.2 | **86.7** |
| | SimCLR(-) | 0 | ✗ | 88.6 | 90.7 | 79.1 | 90.0 | 90.1 | **91.1** |
| | | 0 | ✓ | 91.2 | 92.9 | 87.3 | 92.5 | 92.6 | **93.0** |
| | | 0.1 | ✗ | 71.4 | 79.6 | 83.4 | 80.8 | 80.2 | **86.3** |
| | | 0.1 | ✓ | 77.9 | 83.9 | 87.2 | 83.6 | 83.1 | **87.8** |
| C100 | BYOL | 0 | ✗ | 78.6 | 79.6 | 49.4 | 74.8 | 76.8 | **81.3** |
| | | 0 | ✓ | 81.0 | 80.2 | 59.5 | 77.5 | 78.2 | **83.4** |
| | | 0.1 | ✗ | 74.1 | 76.2 | 50.8 | 71.4 | 73.4 | **77.7** |
| | | 0.1 | ✓ | 78.0 | 77.8 | 59.5 | 75.0 | 75.7 | **80.7** |
| | SimSiam | 0 | ✗ | 79.7 | 81.4 | 55.6 | 77.1 | 78.8 | **83.3** |
| | | 0 | ✓ | 84.5 | 84.3 | 72.1 | 82.7 | 83.1 | **86.6** |
| | | 0.1 | ✗ | 73.1 | 78.9 | 54.7 | 75.4 | 75.4 | **79.7** |
| | | 0.1 | ✓ | 79.7 | 80.3 | 67.2 | 78.4 | 78.4 | **82.5** |
| | SimCLR | 0 | ✗ | 76.3 | 77.0 | 35.5 | 76.3 | 78.2 | **84.2** |
| | | 0 | ✓ | 80.1 | 82.0 | 68.4 | 82.6 | 82.7 | **86.9** |
| | | 0.1 | ✗ | 69.2 | 75.9 | 46.5 | 75.0 | 73.8 | **79.9** |
| | | 0.1 | ✓ | 75.8 | 80.0 | 65.6 | 79.2 | 78.6 | **83.0** |
| | SimCLR(-) | 0 | ✗ | 83.4 | 84.7 | 68.5 | 84.9 | 85.9 | **87.9** |
| | | 0 | ✓ | 85.8 | 87.0 | 80.3 | 87.4 | 87.8 | **89.4** |
| | | 0.1 | ✗ | 78.1 | 80.5 | 67.3 | 81.1 | 81.0 | **84.3** |
| | | 0.1 | ✓ | 81.2 | 82.8 | 79.5 | 83.9 | 83.6 | **85.8** |
| fMNIST | BYOL | 0 | ✗ | 90.5 | 95.3 | 84.7 | 95.0 | 95.4 | **95.9** |
| | | 0 | ✓ | 93.2 | 95.1 | 91.2 | 94.8 | 95.0 | **96.2** |
| | | 0.1 | ✗ | 38.1 | 61.3 | 84.9 | 75.5 | 75.2 | **86.7** |
| | | 0.1 | ✓ | 48.0 | 73.2 | 86.9 | 80.9 | 80.9 | **87.9** |
| | SimSiam | 0 | ✗ | 92.7 | **95.9** | 84.7 | 95.7 | 95.8 | 95.8 |
| | | 0 | ✓ | 93.9 | 95.0 | 90.7 | 94.8 | 94.9 | **96.1** |
| | | 0.1 | ✗ | 40.3 | 63.0 | **88.4** | 73.3 | 72.6 | 86.1 |
| | | 0.1 | ✓ | 52.7 | 75.3 | **90.3** | 80.0 | 79.8 | 87.8 |
| | SimCLR | 0 | ✗ | 87.6 | 94.6 | 70.4 | 95.0 | 95.1 | **96.1** |
| | | 0 | ✓ | 91.3 | 94.9 | 86.8 | 94.9 | 95.0 | **96.3** |
| | | 0.1 | ✗ | 30.9 | 46.5 | **88.4** | 55.5 | 55.0 | 86.3 |
| | | 0.1 | ✓ | 34.9 | 53.1 | **90.6** | 61.1 | 60.6 | 87.5 |
| | SimCLR(-) | 0 | ✗ | 92.7 | 94.7 | 86.3 | 94.5 | 94.5 | **95.6** |
| | | 0 | ✓ | 94.2 | 95.7 | 90.4 | 95.6 | 95.6 | **95.9** |
| | | 0.1 | ✗ | 60.9 | 78.7 | **90.5** | 83.7 | 83.5 | 90.4 |
| | | 0.1 | ✓ | 65.3 | 80.9 | **92.1** | 84.4 | 84.3 | 90.9 |

## H.2 Self-supervised learning (SSL)

SSL is a form of unsupervised learning, tackling it through means of supervised learning from pseudo-labels that can easily be generated. One line of computer vision research uses augmentations as pseudo-labels. These augmentations can be generated at no additional human cost, for example 90 degree rotations results in four labels. In jigsaw tasks (Noroozi & Favaro, 2016) the image is split in grids, for example 2x2 or 3x3, and shuffled, the resulting position is the prediction target.

Another more recent direction is constrastive learning (Oord et al., 2018; He et al., 2019; Misra & van der Maaten, 2019; Grill et al., 2020a). In SimCLR (Chen et al., 2020a;b) every image in a batch is augmented twice, and the objective is to minimize the distance of the latent representations of the same origin image, while maximizing the distance to other images in the batch.

Another recent SSL direction is non-contrastive or positive samples only SSL. Bootstrap Your Own Latent (BYOL) (Grill et al., 2020b) was the first example of this class of algorithms to achieve very competitive results, that even surpasses SimCLR. BYOL gets away from the problem of representation collapse (first enemy of SSL, usually handled by negative samples) by introducing assymetry in the network architecute through the idea of a prediction network after the project head, it also uses an exponential moving average of the weights of the network as a target representation. SimSiam (Chen & He, 2020) made a significant analysis on BYOL and found that using a moving average of the weights wasnot necessary and just a simple `stop-grad` operation was enough.

