# OpenReview forum: "No Shifted Augmentations (NSA): strong baselines for self-supervised Anomaly Detection"
_ICLR.cc/2022/Conference — ICLR 2022 Submitted_

### Official Review · Reviewer_p72E · 2021-11-01

**Correctness:** 4
**Technical Novelty And Significance:** 4
**Empirical Novelty And Significance:** Not applicable
**Recommendation:** 6
**Confidence:** 3

**Main Review:**

+ (+) The paper introduces several insightful contributions: e.g. AD correlation with distribution uniformity measured by fitting a vMF distribution or by MMD; or solution to the gradient-flow issue in SimSiam.
+ (+) The proposed improvements to SSL anomaly detection do not need to make assumptions on the data invariances
+ (+) The proposed multi-scale feature ensamble is an effective way to exploit cues at different semantics/resolutions without increasing inference time
+ (+) The proposed improvements are claimed to hold even under mild data pollution

MAJOR

+ (-) The paper is not well organized: starting from Sec. 2.2 going forward the reader is continuously referred to the appendix to see plots, tables and details that the reviewer finds essential to understand, appreciate and validate the authors' claims.

MINOR

+ typos: ood evaluation -> OOD evaluation (Sec 1, page 1); to actually to decrease -> to actually decrease (Sec. 2.3 page 5);
+ Table B is references but it does not exists in the paper nor in the appendix.

**Summary Of The Paper:**

The paper contributes to the field of self-supervised anomaly detection by i) verifying the correlation between AD performance and in-distirbution (ID) representation uniformity and how to improve it with a simple normalization; ii) introducing a simple solution to a gradient-flow issue in previous work; and iii) proposing a multi-level feature ensembling as a cheaper but quantitatively effective alternative to test-time data augmentation and model ensambling. They build on SimSiam and SimCLR to support all of their claims.

**Summary Of The Review:**

On one hand the paper has good contributions, provides insights into the problem that are supported by empirical evidence; on the other hand is poorly organized and written: most of the key results are supported in the appendix, which should be optional - not critical - to the reading of the paper.

---

> ### Author Response · Authors · 2021-11-24
> **Answers to reviewer p72E's questions**
>
> We would like to thank the reviewers for his comments, and very good summary of the strengths of the paper
>
> > The paper is not well organized:
>
> The paper went through a massive revision in its organization, placement of tables and figures, and exposition/explanation of the core contributions, ideas, and analysis.
>
> > typos..., Table B is references but it does not exists
>
> These are all fixed now. Thank you for pointing these out.

---

### Official Review · Reviewer_9Mbn · 2021-11-02

**Correctness:** 3
**Technical Novelty And Significance:** 2
**Empirical Novelty And Significance:** 3
**Recommendation:** 6
**Confidence:** 3

**Main Review:**

Strengths:

(1) This work looks solid, and the experiments in this paper are sufficient, which consider almost all key parameters or components of the proposed method and demonstrate the effectiveness of the method. Besides, the method achieves state-of-the-art performance on OOD task.

(2) The paper spots the reliability issue on long term training of the contrastive Self-Supervised Learning (SSL) methods as deep feature extractors on OOD. The authors claim that no-uniformity and compactness of the learned ID distribution is the main reason to improve the performance on OOD.





Weaknesses:

(1) The proposed method seems simple, the proposed NSA is built on a modified (l2-norm) SimSiam architecture.

(2) The title states that authors targets on anomaly detection, but this paper works on out-of-distribution detection. They are not the same problem: the ID for anomaly detection is one-class; and that for out-of-distribution detection is multi-class.

(3) This manuscript is not very well organized. The organization of tables, figures and text is in a mess. Tables and Figures shall not be placed between text, and the distance from the table to its reference is somehow long. For example, Table 6 should not be place behind the conclusion, which should be the end of the paper. And the paper exceeds the 9-page limit.





Improvements:

(1) It would be better to combine Figure3 and Figure 7, Figure 4 and Figure8. The organization of figures makes readers hard to follow.

(2) I wonder the slight difference between with and without normalization showed in Figure 2, especially on SimCLR (the same as Figure 4 and Figure8). In addition, Figure 2(b) miss to show the result of SimCLR w/o norm.

(3) The equation in Figure 1 is l1-norm, but the paper states to employ l2-norm. Figure 1 is the most important figure in this paper. This confuses me a lot.


**Summary Of The Paper:**

In this paper, authors study anomaly detection under fully unsupervised setting. This work explores novel architectural modifications on SimSiam (Chen & He, 2020) to the self-supervised feature learning step, that facilitates compact in distribution (ID) distributions to be learned on out-of-distribution (OOD) detection. It performs robustness to polluted ID. In addition, the paper investigates how the geometrical compactness of the ID feature distribution makes isolating and detecting outliers easier.

**Summary Of The Review:**

Overall, the paper presents some good empirical results of OOD detection on benchmarks. This paper is well motivated and solid, but the writing needs to be improved. However, it is somehow weak in terms of technical novelty. Therefore, I vote this paper marginally below the acceptance threshold.
Given that I am not very familiar with OOD, I would like to check other reviewers' opinions.

---

> ### Author Response · Authors · 2021-11-24
> **Answers to reviewer 9Mbn's questions**
>
> We would like to thank the reviewer for their detailed comments and suggestions.
>
> > The proposed method seems simple, the proposed NSA is built on a modified (l2-norm) SimSiam architecture.
>
> We’d like to appeal to Occam’s razor here. This comment about the simplicity of the method is listed under weaknesses. We would like to argue that given the novelty of proposing this normalization for inducing a compact distribution and the extensive empirical studies that demonstrates the method’s effectiveness, the simplicity of our method should be considered a strength, as was also stated by Reviewer ft5T.
>
> Simpler methods are easier to understand, easier to implement, easier to debug, and often easier to maintain. Indeed, as the reviewer points out, “This work looks solid”, “effective” and “the method achieves state-of-the-art performance on OOD task” - all of which we would argue are better to achieve with a simple method than an overly elaborate one that is likely more brittle.
>
> > The title states that authors targets on anomaly detection, but this paper works on out-of-distribution detection. They are not the same problem: the ID for anomaly detection is one-class; and that for out-of-distribution detection is multi-class.
>
> We agree with the point the reviewer is making about the delicate distinction between Anomaly detection (AD) and OOD detection. However, in all our evaluations in the paper, all inlier (ID) data is treated as a single class, and outlier (OOD) data the other classes from the same dataset, which makes it closer to AD. We would like to also point out that the clear cut difference between “classes” and “datasets” may not always be attainable in many real world settings and thus both can be equally correct in a practical (non-benchmark) setting.
>
> To be clear, when we state “OOD” we are referring in a general sense to outlier data for whatever kind of experiment we are considering, as opposed to the inlier (ID) data, and not “OOD detection” which is sometimes used in prior works to refer to the multi-dataset case. We agree the terminology is a little confusing and we have tried to clarify our intent.
>
> > This manuscript is not very well organized.
>
> This is a very fair point and we have tried and believe we have succeeded in fixing this issue. This current version has undergone a massive revision in its organization, placement of tables and figures, and exposition/explanation of the core contributions, ideas, and analysis, taking into account feedback from all reviewers.
>
> > It would be better to combine Figure3 and Figure 7, Figure 4 and Figure8. The organization of figures makes readers hard to follow.
>
> Agreed. We now have all these figures next to each other in the appendix, and a summary figure in the main text combines everything into one graph for each type of experiment, showing both with and without our proposed normalization.
>
>
> > I wonder the slight difference between with and without normalization showed in Figure 2, especially on SimCLR (the same as Figure 4 and Figure8). In addition, Figure 2(b) miss to show the result of SimCLR w/o norm.
>
> We now have with and without normalization on the same graph, so that differences are much easier to see and note.
>
> > The equation in Figure 1 is l1-norm, but the paper states to employ l2-norm. Figure 1 is the most important figure in this paper. This confuses me a lot.
>
> Thanks for pointing out, we now have made it much clearer on the figure that L2-norm is used everywhere.

---

> > ### Comment · Reviewer_9Mbn · 2021-11-29
> > **Response**
> >
> > Thank you on careful discussion.
> >
> > I carefully went over the other reviewers' reviews and the authors' responses, and found that it does not fully resolve some of my initial concerns. I still feel that it is a borderline paper - I changed my decision to borderline accept.

---

> > > ### Author Response · Authors · 2021-11-29
> > > **Reply to reviewer 9Mbn**
> > >
> > > We would like to thank the reviewer for re-evaluating the paper and raising their score in response!
> > >
> > > > Indeed, the added normalization is not so novel for me.
> > >
> > > We would like to highlight an important point here, although the mathematical normalization operation itself is not novel, we are confident that our proposal of using normalization during training for inducing a more compact distribution that much better suits OOD detection is novel, we are unaware of any previous work proposing or demonstrating this. Not only do we show it works across many situations, we also carry a deep analysis demonstrating why it works, and what specific effect it brings to the learned distribution.
> > >
> > > Other novel aspects of the paper include:
> > > - Discovering and solving a previously un-reported gradient flow problem in BYOL/SimSiam. Given how popular BYOL/SimSiam is (in general, outside of Anomaly Detection), we think this is an important point.
> > > - Proposing a feature ensembling mechanism for OOD that is both effective and computationally efficient.
> > > - Demonstrating very competitive OOD performance is attainable without negative augmentations
> > > - Demonstrating a catastrophic failure case of pre-trained models used for OOD (SVHN)
> > >
> > > >  found that it does not fully resolve some of my initial concerns.
> > >
> > > It would be great if you would mention specific points, we would love to take time and clarify our point of view about anything not fully resolved. thanks!

---

### Official Review · Reviewer_ft5T · 2021-11-03

**Correctness:** 3
**Technical Novelty And Significance:** 2
**Empirical Novelty And Significance:** 2
**Recommendation:** 5
**Confidence:** 5

**Main Review:**

- Strength
  - The paper made a good presentation with motivating empirical results to justify their proposed method.
  - The proposed method is rather simple, but shown to be effective.

- Weakness
  - Ablation study on the proposed Mahalanobis cosine similarity is missing. I wonder they are really the necessary component that improves upon existing one class classifiers proposed by [Lee et al., 2018b](https://proceedings.neurips.cc/paper/2018/file/abdeb6f575ac5c6676b747bca8d09cc2-Paper.pdf) or those studied in [Sohn et al., 2021](https://openreview.net/forum?id=HCSgyPUfeDj).
  - [Sohn et al., 2021](https://openreview.net/forum?id=HCSgyPUfeDj) suggested to use L2 normalized encoder output for one-class classification. For models without normalization, I wonder whether authors use L2 normalized encoder output or unnormalized ones.
  - The motivation for adding L2 normalization layer at the end of the projection layer of SimSiam in Section 2.4 needs clarification. It is unclear what it means by "all the gradients of that operation gets blocked by the stop-grad, though it is an integral ...". Also, there seems no specific ablation study provided (by default, I assume "w/ norm" model is with L2 normalization both at the end of encoder and projection layer and "w/o norm" with no L2 normalization layer at all).
  - While mentioned multiple times, there is no empirical result using BYOL, so it is unclear their behavior will be also similar to SimSiam.
  - I would be a bit careful when saying "Feature ensembling does not require a validation set". It is rather heuristic and has been shown (e.g., [Defard et al., 2021](https://arxiv.org/pdf/2011.08785.pdf) or [Rippel et al., 2021](https://ieeexplore.ieee.org/stamp/stamp.jsp?tp=&arnumber=9412109)) specifically for anomaly detection that one should be careful to select correct layers to aggregate.
  - Feature ensembling could be effective in practice, but it makes comparison to other methods somewhat convoluted, especially when readers want to understand the source of improvement. It would be beneficial to have results using only encoder output in Table 1 or Table B.

- Misc
  - For Figure 3 ~ 8, legends are not properly cited.
  - Mahalinobis -> Mahalanobis.


**Summary Of The Paper:**

The paper studied the issues of recent self-supervised contrastive learning based anomaly detection and presented few modification to resolve those issues. Specifically, the paper pointed out the issue of contrastive learning for OOD detection when trained longer as their embedding distributions become more uniform, and proposed an architectural modification by adding a L2 normalization layer after encoder. While SimSiam, a non-contrastive self-supervised learning algorithm, suffers less from uniformity issue, authors found adding L2 normalization layer to SimSiam effective. Authors also pointed out some gradient flow issue of SimSiam and proposed adding another L2 normalization layer at the end of projection layer. Evaluation on anomaly detection verify the importance of the proposed L2 normalization layers, but the per

**Summary Of The Review:**

- The technical contribution of the paper is very simple (adding few L2 normalization layers). While showing some good empirical evidences, the overall contribution of the paper is marginal and does not extend state-of-the-art. Also, while proposing many small techniques, they are not fully verified in experiments (please see my comment above).

---

> ### Author Response · Authors · 2021-11-24
> **Answers to reviewer ft5T's questions**
>
> We would like to thank the reviewer for their detailed comments, and effort reading the paper.
>
> > Ablation study on the proposed Mahalanobis cosine similarity is missing
>
> The new version of the paper provides a very detailed ablation comparing 6 different metrics, one of them is the Mahalanobis cosine, in Appendix G, where we show that for example compared to GDE used in previous works as a scoring function, we gain consistent improvements.
>
>
> > Sohn et al., 2021 suggested to use L2 normalized encoder output for one-class classification. For models without normalization, I wonder whether authors use L2 normalized encoder output or unnormalized ones.
>
> We use L2 normalized outputs as an input for the evaluation metric for all models. This is by definition for most of our functions as they are based on cosine similarity, for the rest (e.g. GDE) we also use L2 normalized output before evaluation
>
> > The motivation for adding L2 normalization layer at the end of the projection layer of SimSiam in Section 2.4 needs clarification. It is unclear what it means by "all the gradients….
>
> We tried to rephrase it in the new version. More informally, if you look at Figure 1, the block computing $\hat {z}_j$ on the far right, we can see that because of the stop-grad operation there are no gradients (red arrows in the figure) flowing from it to other blocks in the network: This is problematic, as this is a part of the network that affects the loss function, but doesn’t produce gradients, so it is not optimized for in the vanilla version of SimSiam.
>
> > Also, there seems no specific ablation study provided (by default, I assume "w/ norm" model is with L2 normalization both at the end of encoder and projection layer and "w/o norm" with no L2 normalization layer at all).
>
> In the new version of the paper, Table 3 provides a detailed ablation covering this specifically for both BYOL and SimSiam on both CIFAR10 and CIFAR100.
>
> > While mentioned multiple times, there is no empirical result using BYOL, so it is unclear their behavior will be also similar to SimSiam.
>
> In the new version of the paper, we included BYOL results in tables 1 and 2, and Appendix G provides a very detailed ablation evaluating BYOL on all setups considered on the paper.
>
> > I would be a bit careful when saying "Feature ensembling does not require a validation set". It is rather heuristic and has been shown (e.g., Defard et al., 2021 or Rippel et al., 2021) specifically for anomaly detection that one should be careful to select correct layers to aggregate.
>
> This is a valid concern, That is why in the new ablation in Appendix G we provide ensembling results on all considered scenarios (SSL models, dataset, and pollution ratios) and empirically verify that the proposed normalization works well across all these scenarios and helps a lot in most situations.
>
> We would also like to point out that, as described in more detail in Appendix C, for our “Ens.” variants of our results in the main text, different layers are handled differently i.e. we use k-Cos for 2D/convolutional maps and c-Cos for 1D/linear maps, and normalize all scores before summation. This combination empirically performed the best on average across all scenarios mentioned above.
>
> > Feature ensembling could be effective in practice, but it makes comparison to other methods somewhat convoluted, especially when readers want to understand the source of improvement. It would be beneficial to have results using only encoder output in Table 1 or Table B.
>
> Unless otherwise stated, all our results presented in the paper are based on the k-Cos metric on the final layer output (no ensembling). When ensembling is used this is now made very clear beforehand. We again refer the reviewer to the new ablation to get a good idea of what is happening and the source of improvement in each case.

---

> > ### Comment · Reviewer_ft5T · 2021-11-30
> > **Response to author**
> >
> > Thanks authors for their response to resolve several of my concerns. I believe they are good (necessary) additions to the paper as the paper involves many small techniques to make it work as comparable as state-of-the-art methods. While the proposed method improve upon its own baseline, I didn't find it compelling as there seems not much qualitative or quantitative benefits over existing methods, so not sure why I should use the proposed methods over the other SOTA methods. I would keep my rating.

---

> > > ### Author Response · Authors · 2021-11-30
> > > **Reply to reviewer ft5T**
> > >
> > > >Thanks authors for their response to resolve several of my concerns. I believe they are good (necessary) additions to the paper
> > >
> > > Thanks for acknowledging that we have resolved many of your concerns, and for acknowledging the quality of our clarifications.
> > >
> > > >While the proposed method improve upon its own baseline
> > >
> > > We would like to bring to the reviewer’s attention that we do improve upon and compare to **all** other state-of-the-art published work in all standard benchmarks we examine. This can be seen in Tables 1,2,4, 5, 10, 11.
> > >
> > > If there is some specific related work we aren’t comparing to, it would be very helpful if the reviewer can specifically point to it.
> > >
> > > > there seems not much qualitative or quantitative benefits over existing methods, so not sure why I should use the proposed methods over the other SOTA methods.
> > >
> > > Quantitative benefits can be seen in our comparisons in Tables 1, 2, 4, 5, 10, 11.
> > > - In presence of pollution, we improve upon state-of-art in all scenarios, manytimes (e.g. C100 and fMNIST) with a **big** performance gap.
> > > - Otherwise, we are at least equivalent to state-of-art
> > >
> > > Qualitatively:
> > > - We can equate state-of-art **without** using negative data augmentations. This is a big advantage, as the assumption of data invariance to these augmentations is a weak assumption, and when the dataset violates it, there is a big drop in performance when using negative augmentations, this is intuitive and was experimentally shown in section 3.2 of [1] (Data-dependence of shifting transformations)
> > > - We can equate or surpass state-of-art ensembling at **much** less computational budget, this can be seen in Table 2
> > >
> > > [1] Jihoon Tack, Sangwoo Mo, Jongheon Jeong, and Jinwoo Shin. CSI: Novelty detection via contrastive learning on distributionally shifted instances. Advances in Neural Information Processing Systems, 2020.

---

### Official Review · Reviewer_UoZi · 2021-11-04

**Correctness:** 4
**Technical Novelty And Significance:** 2
**Empirical Novelty And Significance:** 3
**Recommendation:** 5
**Confidence:** 4

**Main Review:**

__-=-Pros-=-__

 * __Problem Significance:__ Using unsupervised learning as a step towards totally unsupervised AD is an approach that is sensible and worth investigating. The problem identified by the authors makes sense and solving it by finding useful unconcentrated representations is reasonable and could be of some general interest outside of deep AD as well.
* __Approach:__ The authors provide a sensible line of reasoning for their approach and it doesn't seem ad-hoc. This point is meant to contrast a paper whose approach seems to be developed through shotgun testing and doesn't provide much insight to why an approach works.
* __Experimental Setup:__ Several different approaches using standard datasets and setups. Reasonable competitors. Small novel contribution with the vBM distribution approach
* __Experimental Results:__ Results strongly indicate that the proposed method does improve the representation for AD.
* __Clarity:__ Generally good.

__-=-Cons-=-__
 * __Novelty:__ The proposed change is quite small. The value of spherical representations has been observed quite a few times, especially in contrastive learning, so inserting it into a few more layers isn't really making a contribution that will likely be broadly useful.
 *__Lack of Theoretical Analysis:__ If this paper would be very strong had a bit of relevant theoretical analysis. For example analyzing the distribution of an optimal representation for some very simple problem setting, perhaps even showing theoretically that its not uniform. Analysis of the form of "Understanding Contrastive Representation Learning through Alignment and Uniformity on the Hypersphere" (Wang & Isola 2020) would be great.
 *__Experimental Results/ Competitor Approaches/ Missing Citations:__ The proposed method is still outshone by other deep AD methods. In particular [outlier exposure](https://openreview.net/forum?id=HyxCxhRcY7) (OE) methods work very well as well as transfer-learning based methods, and [combining the two](https://proceedings.mlr.press/v139/deecke21a.html) achieves 99%+ AUC on your CIFAR10 experiments. This is somewhat counter to the assertion that "the assumption...that learned image features from general datasets such as ImageNet will transfer well in all cases may be restrictive at best." I think the non-OE non-transfer learning based methods are interesting in their own right, but I think these other approaches/results should at least be mentioned.
 *__Some parts could use more detail:__ The paper should really include a precise description of SimSiam and SimCLR since they are so integral to the contribution you are making. BYOL should also be introduced (also the full name of BYOL should be mentioned somewhere instead of going directly to the acronym). This should be in the appendix, at the very least. "Mahalanobis space" isn't particularly standard or common nomenclature and should be described precisely somewhere (in the appendix if necessary).
* __Quite a few style and grammar errors:__ At end.

__-=-Verdict-=-:__ As is the paper is just a bit worse than what I would expect out of ICLR, however there is a lot of easy room for improvement.


__-=-Other Misc. Errors-=-__
* __Bibliography:__ There are numerous errors in here and the author should go through all the cited works carefully. Below are some examples
  * There are several citations that reference arXiv version despite the existence of peer-reviewed and/or archival versions. I only checked two arXiv works but they bot had peer reviewed archival versions (Ruff 2020 has a [workshop version](https://github.com/lukasruff/Classification-AD), Tack 2020 has a [NeurIPS paper](https://proceedings.neurips.cc/paper/2020/hash/8965f76632d7672e7d3cf29c87ecaa0c-Abstract.html) ).
  * Sehwag 2021 is missing a venue entirely.
  * Tack 2020 "Csi" should all be capitalized, you can do this with {CSI}
  * Schölkopf 1999, I'm not sure what is standard procedure now, but you may want to consider using "NeurIPS" since there [is some controversy surrounding its old name)[https://www.nature.com/articles/d41586-018-07476-w].
* __Capitalization:__ There are quite a few random and incorrect capitalized words. Examples
  * p.1 "Deep Neural Networks" should be "Deep neural networks"
  * p.1 "Industrial" should be "industrial"
  * p.1 "Self-Supervised Learning" should be "Self-supervised learning"
*__Figures in weird locations__: Several of the figures referred to in the main text are in the appendix. The main text should really be self-contained, so you should at least give a summary of the figure in the main text and treat the figures in the appendix as additional information.
* __Some figure keys are too small:__ Figure 2 right is the worse offender of this
* __Figure 2 caption:__ y axis label overlaps the graph, should contain a brief description of kappa and MMD
* __Small errosr:__
  * p. 7: bottom missing space in "Section2.3"
  * p. 6: Not sure what is meant by "high" and "low" level scores
  * " are two random training and test samples.." should be "are a training and test sample respectively"
  * MMD should be introduced with full term (Maximum Mean Discrepancy) and briefly described
  * "use the tool of the von Mises-Fisher" shouldn't use the word "tool" in my opinion, its just a distribution.
  * Figure 1: should use \left< foo \right> to get correct angle brackets
  * (1): replace comma with period

**Summary Of The Paper:**

In this paper the authors propose a way to improve the representation learned from contrastive learning when the downstream task is anomaly detection/out of distribution detection (AD), specifically an unsupervised AD problem where one _only_ uses the unlabelled training data which is assumed to contain only, or mostly, nominal data. Several popular contrastive learning methods map samples to points on a unit hypersphere and find a representation that encouraged to be uniform over the sphere. This is problematic for AD since one needs the nominal data to be concentrated so that it is clear when a test sample lies away, thereby being indicative of an anomaly. To remedy this issue the authors enforce the representation at multiple levels to be on the unit sphere (they look at SimSiam and SimCLR in particular) thereby making it impossible to enforce the contrastive similarity loss to use scale as a way to measure similarity, it must always use direction. This is demonstrated to be effective and makes representations which are more concentrated, useful for AD, and are more robust to pollution and varied batch sizes.

**Summary Of The Review:**

Paper is borderline, could be improved significantly by improving the exposition. I've included several suggestions for this.

---

> ### Author Response · Authors · 2021-11-24
> **Answers to reviewer UoZi's questions**
>
> We would like to thank the reviewers for his detailed comments, and helpful suggestions!
>
> > Lack of Theoretical Analysis: If this paper would be very strong had a bit of relevant theoretical analysis. For example analyzing the distribution of an optimal representation for some very simple problem setting, perhaps even showing theoretically that its not uniform.
>
> We think that a grounded theoretical framework rigorously proving that the added normalization prevents converging into a uniform distribution would be a very interesting future direction. However, we think that the current work provides extensive and sufficient empirical benchmark evidence supporting the practical benefits. Also, our extensive accompanying empirical analysis showing the correlation between AD performance (AUROC scores under different conditions and epochs) and non-uniformity / compactness of the representations’ distributions gives a sound framework for what is actually achieved by the added normalization.
>
> We also note that previous work [1] considered empirical analysis showing that uniformity was harmful for AD via considering only the MMD metric from uniformity at the final learned representation, whereas we consider its evolution during training, and further investigate using the vMF distribution which provides further insight.
>
> [1] Kihyuk Sohn et al. Learning and evaluating representations for deep one-class classification. ICLR 2021
>
>
> > Novelty: The proposed change is quite small. The value of spherical representations has been observed quite a few times, especially in contrastive learning, so inserting it into a few more layers isn't really making a contribution that will likely be broadly useful.
>
> We think there might be a mis-understanding here, we are not just introducing a spherical representation into an intermediate layer, and finding it useful. Rather, we show how to make the learned representation of the convolutional encoder non-uniform, and thus more suitable for OOD. We empirically demonstrate the non-uniformity of the learned representation through (i) computing MMD to a uniform distribution and (ii) estimating kappa of a vMF.
>
> In contrastive learning, spherical representations were found useful for facilitating uniform distribution on a sphere, which promotes easier separability of in-domain data; we here try to demonstrate the exact opposite, how to learn a compact and non-uniform representation, that would be easier to separate, as a whole, from the open world.
>
> We believe that this is a novel use of normalization, and aren’t aware of other cases in literature, where normalization is used specifically for inducing compactness.
>
> > Experimental Results/ Competitor Approaches/ Missing Citations: The proposed method is still outshone by other deep AD methods....
>
> We would like to point to a few things
> * We originally had a coverage of transfer learning based methods, it is been extended now, results from pretrained models can be found in new manuscript in Tables 4,5, and 10;
> * Transfer learning based methods enjoy access to very large amount of external labeled data (that may contain overlap with classes that should be assumed to be “unseen” in the outlier data). It would be unfair to compare these models directly to models like ours that learn from scratch.
> * Transfer learning methods only work well on classes similar to (and containing discriminative features for learning suitable representations to) those on which the base model was pre-trained;
>     * We show in Section 3.2.5 that the best published transfer learning models all fail on the simple SVHN;
>     * In a parallel ICLR submission (https://openreview.net/forum?id=vruwp11pWnO) authors show more results demonstrating bad generalization by pre-trained models. We second this view that evaluating ImageNet pre-trained models on CIFAR 10 or CIFAR 100 is evaluation on the training set.
>
> * We didn’t include OE methods in comparisons as we feel they are out of scope for our work. The OE’s assumption of the availability of a large amount of data with high similarity to OOD data is not practical for many situations. Finally, we think it would be an interesting follow-up work to evaluate proposed methods in the OE scenario.
>
> > Some parts could use more detail:....
>
> Thanks for noticing this, we now have Appendix H, which discusses all related work, and more background on the SSL methods we build upon. “Mahalanobis space” is now described mathematically in the main paper.
>
> > Quite a few style and grammar errors: At end.
>
> > Other Misc. Errors
>
> We really appreciate the effort taken by the reviewer to extract and point to all this! All now fixed in the new version

---

### Author Response · Authors · 2021-11-24
**Answers to all reviewers**

Thank you for reviewing our paper and providing insightful comments. We carefully read all the reviews and provided detailed answers in the replies below each review. We in particular are grateful to the reviewers for their feedback on the paper presentation and organization which was a common theme, and we have attempted to rectify and significantly improve this, taking these suggestions into account.

First of all we summarize that the reviewers recognized that we:
* "*provide a sensible line of reasoning*"; address a significant problem that "*could be of some general interest*" outside this field; has results that "*Strongly indicate*" we "*improve the representation for AD*" [UoZi]
* Has "*good presentation with motivating empirical results*", is "*simple but effective*" [ft5T]
* achieve state-of-the-art results; our work is "*solid*" and with "*sufficient experiments*"; we "*spot the reliability issue*" with long term contrastive SSL training [9Mbn];
* "*introduces several insightful contributions*"; in contrast to other methods we "*do not need to make assumptions on the data invariances*"; we demonstrate advantages in ensembling "*without increasing inference time*" [p72E].


We consider our revision builds on these strengths. We would nevertheless like to make the following points in general in response to the suggestions for improvement:

- Our *full source code* is now available here: https://anonymous.4open.science/r/nsa-E291

- **Novelty/significance vs simplicity**: We would like to stress that while the approach is simple, it’s novelty and significance is substantial. In fact, we consider the simplicity of our approach as a strength - it is as complicated as it needs to be, and no more. We see similar effects in some important other work in deep learning - DropOut, BatchNorm, skip-connections, ReLU, and momentum SGD etc can all be considered as simple modifications of previous work but they have significant impact in their respective areas.

- We uploaded a new version of the manuscript, with the following highlighted modifications:

    * The paper went through a massive revision in its organization, placement of tables and figures, and exposition/explanation of the core contributions, ideas, and analysis.

    * A summary figure is now present in the main body of the paper, while detailed figures are in the appendix. We’ve done the same for the tables.

    * A very detailed ablation study is now included in Appendix G, which includes results of training 640 different models and evaluating each model using 6 different metrics, including the proposed feature ensemble.

    * We now fully evaluate BYOL, SimSiam, SimCLR, and SimCLR with negative augmentations on CIFAR10, CIFAR100, and f-MNIST in presence and absence of pollution.


We hope all reviewers can take these revisions and improvements into account and consider the improvements we have made in line with their recommendations.

---

### Author Response · Authors · 2021-11-27
**Request for discussion**

Dear reviewer,

Since the author response period is closing very soon (Nov 29th), we wanted to send a gentle request for discussion. Please find our responses to the original review below, and the revised paper that we have uploaded. It will be very helpful if you could let us know whether we have addressed your concerns and if we can provide any additional clarification for a revised assessment of our paper.

Thanks, Authors

---

### Decision · Program_Chairs · 2022-01-20

**Decision:**

Reject

**Comment:**

The paper investigates how the geometrical compactness of in-distribution examples affects OOD detection performance and proposes architectural modifications to enable compact in-distribution embeddings. All the reviewers agreed that the paper has several interesting contributions. I agree with the authors that simplicity is a strength, not a weakness.

My main concern is that the paper's contributions feel a bit scattered. For instance, the paper does a detailed evaluation of normalization and compactness, but makes a few other minor contributions (as detailed by
the authors at https://openreview.net/forum?id=7VH_ZMpwZXa&noteId=m-1y5byLbwS​). However, the latter contributions feel a bit narrow to specific methods and are not as comprehensively tested as the claims around normalization.

Overall, the reviewers and I think that the current version falls below the acceptance threshold. I encourage the authors to revise the draft and resubmit to a different venue.